# The land-sea coastal border: A quantitative definition by considering the wind and wave conditions in a wave-dominated, micro-tidal environment

Agustín Sánchez-Arcilla[1], Jue Lin-Ye[1], Manuel García-León[1], Vicente Gràcia[1], and Elena Pallarès[1,2]

[1]Laboratory of Maritime Engineering, Barcelona Tech, D1 Campus Nord, Jordi Girona 1-3, 08034, Barcelona, Spain
[2]EUSS - Escola Universitaria Salesiana de Sarria, Sant Joan Bosco 74, 08017, Barcelona, Spain

**Correspondence:** A. Sánchez-Arcilla (agustin.arcilla@upc.edu)

**Abstract.** A quantitative definition for the land-sea (coastal) transitional area is here proposed for wave-driven areas, based on variability and isotropy of met-ocean processes. Wind velocity and significant wave height fields are examined for geo-statistical anisotropy along four cross-shore transects on the Catalan coast (northwestern Mediterranean) illustrating a case of significant changes along shelf. The variation of the geo-statistical anisotropy as a function of distance from the coast and water depth has been analyzed through heatmaps and scatter plots. The results show how the anisotropy of wind velocity and significant wave height decrease towards the offshore, suggesting an objective definition for the coastal fringe width. The more viable estimator turns out to be the distance at which the significant wave height anisotropy is equal to the $90^{th}$ percentile of variance of the anisotropies within a $100\text{km}$ distance from the coast. Such a definition, when applied to the Spanish Mediterranean coast, determines a fringe of width of $2\text{-}4\text{km}$. Regarding the probabilistic characterization, the inverse of wind velocity anisotropy can be fitted to a lognormal distribution function, while the significant wave height anisotropy can be fitted to a log-logistic distribution function. The joint probability structure of the two anisotropies can be best described by a Gaussian copula, where the dependence parameter denotes mild to moderate dependence between both anisotropies, reflecting a certain decoupling between wind velocity and significant wave height near the coast. This wind-wave dependence remains stronger in the central, bay-like part of the study area, where the wave field is being more actively generated by the overlaying wind. Such a pattern controls the spatial variation of the coastal fringe width.

## 1 Introduction

Land-sea border areas are narrow strips of water that display unique met-ocean dynamics due to a) non-linearity and sea bottom interactions (including bathymetric control) for the ocean (Shaw et al., 2008) and b) differential heat over land/sea and topographic control on winds (e.g. channeled winds and coastal jets) (Miller et al., 2003; Estournel et al., 2003). This results in enhanced gradients that interact with very productive ecosystems and a large number of infrastructures and socio-economic uses related to tourism, fisheries/aquaculture or maritime transport (Halpern et al., 2008; Bulleri and Chapman, 2010; Barbier et al., 2011; Sánchez-Arcilla et al., 2016). However, the limits of this land-sea transition remain fuzzy and even somewhat subjective, depending on the type of process or application considered and with technical, economic and legal implications.

There is, thus, a need for a systematic and objective definition of the coastal fringe that considers underlying processes and that has general applicability allowing for the time/space dynamics of this fringe. This type of approach has been explored in the literature, where for instance Sánchez-Arcilla and Simpson (2002) reviewed a number of possibilities based on a dynamic balance of competing processes (i.e. drivers) such as inertial effects, geostrophic steering, sea bed friction or water column stratification. Another suitable option is to focus on the consequences of such processes, such as the nearshore morphodynamic features (Geleynse et al., 2012) (i.e. deltas, sand spits, overwash fans, beach berms). Both complementary classifications requires spatial data that needs to be updated accordingly within timescales that may range from years (i.e. long-term erosion due to sea level rise) to days (i.e. storm-scale).

In the last decades, the advent of remote sensing has led to enviromental monitoring at spatio-temporal scales hard to achieve previously, with just in-situ measurements. Hence, such high spatial resolution and short revisit time offer an alternative source of information for such a coastal zone definition, although with some limitations since the data may start degrading at a few kilometres (order $10km$) offshore from the coast (Cavaleri and Sclavo, 2006; Wiese et al., 2018; Cavaleri et al., 2018). The land boundaries induce error in the satellite observations. Hence, it is useful to use high resolution numerical simulations supported by in-situ data so that land-sea boundary effects are properly captured for the subsequent coastal definition that will be based on the heterogeneity introduced by the presence of the land boundary.

The geo-statistical anisotropy (henceforth, anisotropy) in wind and wave fields (Swail et al., 1999) can be an useful indicator of spatial structure, affected by topo-bathymetric constraints that generate substantial gradients in met-ocean conditions that are wave-driven. In this text, the term "anisotropy" refers only to the geo-statistical anisotropy, not the geo-physical one. A wind or wave field that has a high anisotropy can present a predominant wind or wave direction, respectively. It is well known that the geo-statistical anisotropy can be a measure to define directional variation e.g. for mineral configuration in rocks (Amadei, 1996), for propagation velocity in heterogeneous media (Crampin, 1984) or for seismic waves (Verdon et al., 2008). Similarly, topographic induced geo-statistical anisotropy affects coastal wind patterns that force wave and current fields (Soomere, 2003).

The aim of this paper is to analyse the geo-statistical anisotropy of nearshore wind and waves, in wave-driven coasts. From that, what follows is to propose a new quantitative and objective definition for the land-sea border that benefits from these high-resolution (spatial and temporal) fields and from the underlying process-based knowledge. This definition can be useful to determine a set of criteria for numerical purposes (e.g. nesting coastal domains) but also for more practically oriented applications (e.g. offshore limit for outfall dispersion). The analysis is based on a set of high-resolution wind and wave fields in the latter case using a well-tested code such as SWAN (Booij et al., 1999; van der Westhuysen et al., 2007; WISE Group, 2007; Zijlema, 2010). The numerical results, pertaining to a micro-tidal environment to avoid any distortion of spatial patterns by tides, will be subject to inexpensive statistical methods to characterize spatial structures. Following this approach, the paper is structured as follows. Section 2 introduces the theoretical background. Section 3 describes the study area, while the methodology is presented in Section 4. Section 5 lists the main results, which are discussed in Section 6, followed by some conclusions in Section 7.

## 2 Theoretical background

Given a spatio-temporal field $X(s,t)$, where $s$ stands for a 2-D vector (zonal and meridional components) and $t$ is the time, it is assumed that the iso-level contours of the correlation functions are invariant, i.e. ellipses in two dimensions. The main axis of these ellipses are termed $u$ and $v$, respectively (see Fig. 1). The metric of the geometric anisotropy, then, becomes their ratio $R = \frac{u}{v}$ ($R \in [0, \infty)$) (Chorti and Hristopulos, 2008; Petrakis and Hristopulos, 2017). An R value close to unity means that $u$ and $v$ are isotropic, i.e. homogeneous across the different directional sectors. As R increases, the difference between the main axis increase, showing higher anisotropy at certain directional sectors.

Considering the ratio $R$ as a 1-D random variable, it can be fit to a probability distribution function. Such fitting depends on theoretical and practical considerations. The preferred shape is determined by looking at statistical characteristics such as mean, variance, skewness and kurtosis, or by examining the similarity between quantiles (dataset versus theoretical probability distribution) using a Quantile-Quantile plot. The more direct candidates to fit variable $R$ are a) the log-normal function, where the probability distribution of its log-transform is Gaussian (Aitchison and Brown, 1957) and b) the log-logistic function, with a logistic probability distribution for the log-transformed variable. A logistic distribution has a probability density function of the form:

$$f(x) = \frac{1}{s} \exp\left((x-m)/s\right)\left(1 + \exp\left((x-m)/s\right)\right)^{-2}, \tag{1}$$

where $m$ is its location parameter and $s$ is its scale parameter.

Sklar's theorem (Sklar, 1959), expresses the multivariate joint probability structure of two variables $x$ and $y$ as the product of their cumulative probability distributions $F(x)$ and $G(y)$, and a 2-D copula. The interval of variation of $F(x)$, $G(y)$ is $[0,1]$ and a 2-D Gaussian copula has the form:

$$C_\rho(F(x), G(y)) = \int_{-\infty}^{\Phi^{-1}(F(x))} \int_{-\infty}^{\Phi^{-1}(G(y))} \frac{1}{2\pi\sqrt{1-\rho_{12}^2}} \exp\left\{-\frac{s^2 - 2\rho_{12}st + t^2}{2(1-\rho_{12}^2)}\right\} \mathrm{d}s\mathrm{d}t. \tag{2}$$

where the correlation parameter $-1 < \rho_{12} < 1$ is used as the dependence parameter and $\Phi$ is the univariate standard normal distribution function (Embrechts et al., 2001). $\rho_{12} = 0$ means total independence between the variables, whereas $|\rho_{12}| = 1$ means total dependence. Whenever a joint probability structure has the form of a Gaussian copula, this structure can be applied without excessive computational cost (Li et al., 2014; Rueda et al., 2016), compared, for instance, to an Archimedean copula approach (Wahl et al., 2011; Lin-Ye et al., 2016; Okhrin et al., 2013).

## 3 Study area

The selected pilot site for application is a wave-driven and micro-tidal environment such as the western Mediterranean Sea (Fig. 2), where enough validated met-ocean simulations exist and where the spatial wind/wave structure will not be distorted

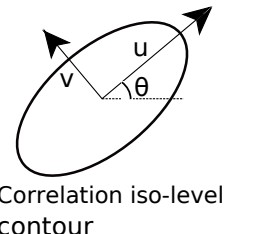

Principal directions

u

v

θ

Correlation iso-level
contour

**Figure 1.** Representation of a generic ellipse that represents the geo-statistical geometric anisotropy of a wind or wave fields. Supposing $u$ and $v$ are the two principal directions of anisotropy, the anisotropy ratio is $R = \frac{u}{v} > 1$. $\theta$ is the rotation angle of the field.

by tidal forcing. Current fields, slower to respond to the overlying meteorological driving, have not been considered in this initial analysis. The focus is on the Spanish north-eastern Mediterranean coast, where we have in-situ and altimeter data for support. Moreover, the continental shelf varies in from 10km to more than 100km in an alongshore distance of less than 500km. The wind fields are affected (most frequent wind direction is from land, corresponding approximately to the north-west) by the
presence of a mountain chain roughly parallel to the coastline and featuring several openings corresponding to river valleys. The geometrical anisotropy analysis has been performed at four transects, characteristic of its common topo-bathymetric features. They correspond (Fig. 2) to the following locations (from south-west to north-east): Ebre (40.7°N, 0.87°E), Tarragona (41.12°N, 1.25°E), Mataro (41.53°N, 2.45°E) and Begur (42.28°N, 3.02°E).

    The north-western Mediterranean presents a particularly intense wind forcing, which is shaped by local orography (Jordi
et al., 2011; Lebeaupin Brossier et al., 2012). The Pyrenees mountain chain across the strip of land connecting the Iberian Peninsula to the European Continent forces a strong northern wind flux following the French-Spanish Mediterranean coast (Nicolle et al., 2009; Schaeffer et al., 2011; Obermann-Hellhund et al., 2017). This same wind pattern is channelled by the river valleys resulting in a north-western orientation for winds blowing from land to sea further down along the coast (Cerralbo et al., 2015), for latitudes southward of $41°N$. The most frequently observed patterns are, thus, from North in the coastal sector
closer to the Pyrenees barrier and from the northwest further south, conditioned by the river valleys and gaps in the coastal parallel mountains (Obermann et al., 2016; Lin-Ye et al., 2016). The second most frequent pattern corresponds to western winds, associated to atmospheric depressions in northern Europe (Barnston and Livezey, 1987; Trigo et al., 2002; Lin-Ye et al., 2017). Easterly winds are frequent during the summer, triggered by an intense high-pressure area over the British Islands.

    The most common wave fields in the north-western Mediterranean Sea correspond to wind-sea (Lionello and Sanna, 2005;
Bolaños et al., 2009) forced by the easterly winds and the northerly and north-westerly winds mentioned above. Because of the semi-enclosed character of the basin, the waves are fetch-limited, with maximum trajectory lengths around $600$km, one-sixth of the average distance that a wave train travels across the Atlantic (García et al., 1993). The average wave-climate in the north-western Mediterranean Sea presents a mean significant wave height ($H_s$) of $0.78$m at the southern part of the Catalan coast, near the Ebre delta and slightly lower values (around $0.72$m) further north and close to the French border. The spatial
distribution for wave storms presents an opposite trend, with maximum $H_s$ between $5.48$m in the southern sector and $5.85$m

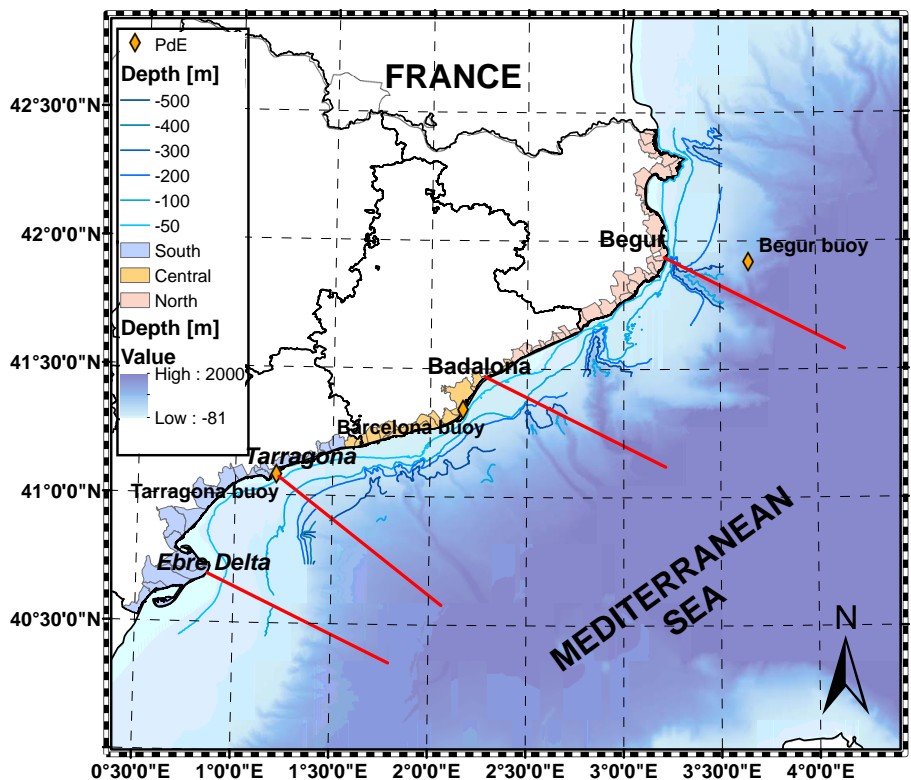

**Figure 2.** Study area, showing the gradients in topo-bathymetry that exert a strong control over the resulting met-ocean conditions. There are depicted the four transects (red lines) used to estimate the limit of the coastal fringe, located in (from South to North): a) Ebre Delta, b) Tarragona, c) Badalona and d) Begur. It also shows the Puertos del Estado (PdE) buoys and the division of the Catalan Coast into the northern, central and southern sections (different colours in the land part).

at the northern coastal stretch (Bolaños et al., 2009). Future projections Casas-Prat and Sierra (2013) indicate, for the interval 2071-2100 and A1B scenario (IPCC, 2000) a variation of the significant wave height around $\pm10\%$, whereas the same variable for a 50year return-period exhibits rates of change around $\pm20\%$. Additionally, the variability in large-scale indices (i.e. NAO, EA or Scandinavian Oscillation) may drive significant changes in wave storm components (Lin-Ye et al., 2017).

## 4 Methods

The approach suggested for assessing the geo-statistical anisotropy of wind and wave fields is schematized in (Fig. 3). It requires high-resolution met-ocean fields to determine how the covariance of the geo-statistical anisotropies of wind and wave fields evolve with distance to the land-sea border. The starting point are wind and associated wave fields, as the suggested candidates for reflecting the heterogeneity induced by coastal topo-bathymetry. Although other definitions of the coastal boundary can be based on river plumes or bio-geochemical processes, it has been intended to focus on a more hydro-dynamical expres-

sion of such boundary for wave-driven coasts. It is intended to show that, as one approaches the coast, the wind and the wave fields should present a greater geo-statistical anisotropy, that is, they should display predominant wind and wave directions. Furthermore, there should be a geo-statistical boundary to the value of this anisotropy that could help define a coastal boundary.

The wind fields have been provided by the UK-Met Office, from their Unified Model (Cullen, 1993) for weather and climate applications. This code solves the compressible, non-hydrostatic equations of motion with semi-Lagrangian advection and semi-implicit time stepping, including suitable parameterizations for sub-grid scale processes such as convection, boundary layer turbulence, radiation, cloud microphysics and orographic drag (Brown et al., 2012). There are two atmospheric prognostics: the dry one (three-dimensional wind components, potential temperature, Exner pressure and density) and the moist one (specific humidity and prognostic cloud fields (Walters et al., 2011). Both long and short radiations (from the sun and the Earth itself) are included, whereas the effect of aerosols reflecting them is taken into consideration. These wind data has been validated in previous works, such as in Martin et al. (2006).

The computational domain of the wind field spans the whole Mediterranean Sea using a regular grid with spacing of 17km and a time step of 1h. The wave fields have been calculated with the SWAN code, covering the same geographical domain and with equal time step of 1h. SWAN is a spectral wave model based on the wave action balance equation (Booij et al., 1999; Zijlema, 2010) that includes non-linear interactions at various depths and dissipation processes (i.e. whitecapping, bottom friction, wave breaking). It applies a fully implicit numerical scheme for propagation in geographical and spectral spaces that is unconditionally stable.

SWAN employs an unstructured grid with spatial resolutions of 600m-40km, denser near the land-sea boundary. Mesh sizes are proportional to bathymetry gradients and distance to the coastline, following the same criteria than in (Pallarés et al., 2017). Such a non-structured grid approach avoids nesting and internal boundary conditions, while maintaining a good spatial resolution to capture bottom and coastline irregularity (submarine canyons and capes or prodeltas that are found in the Catalan continental shelf). Furthermore, unstructured meshes are well suited to tackle non-linear effects (Qi et al., 2009; Roland et al., 2012; Roland and Ardhuin, 2014). The resulting wave fields have been validated with two directional wave buoys at the northern (Begur, deployed at 1200 m) and southern (Tarragona coastal buoy, deployed at 15 m) ends of the domain, managed by Puertos del Estado (Fig 2). Altimeter data from three satellites (Cryosat,Jason-2 and Jason-3) are also used as a complementary observational source. The simulation period ranges from October 2016 to March 2017.

Once obtained the wave outputs, the empirical semi-variograms for the significant wave height and the wind velocity at 10m are estimated. In order to have enough data, the spatial radius of influence is assumed to be 5km, plus time blocks of 24 hours. From these semi-variograms, the anisotropy for the wave height ($R_{H_s}$) and the wind velocity ($R_{V_w}$) is estimated along the four transects in Fig 2.

Distance to the coast ($x$) and water depth ($h$) are selected as independent variables for analysing the anisotropy spatial patterns. $R_{V_w}$ and $R_{H_s}$ are taken to represent the behaviour of met-ocean conditions under the effect of the land-sea boundary (in this initial analysis height/depth gradients in topo-bathymetry). Hence, $R_{V_w}$ and $R_{H_s}$ have been interpolated (1km spacing) along a 100km transect perpendicular to the coast (see Fig. 2), considering periods of 24h, long enough for the waves to respond to the acting wind forcing.

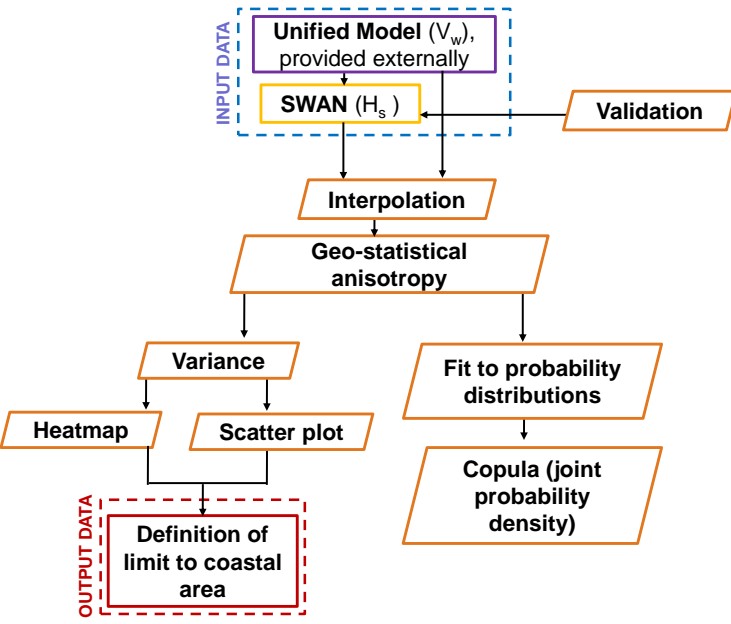

**Figure 3.** Flow-chart summarizing the methodology used in this paper. The dashed blue rectangle represents the input data, the red dashed rectangle indicates the output data. The wind velocity is obtained from an external source, and it was validated in Martin et al. (2006). The rest of the steps have been carried out for this analysis. Rectangles indicate data generation (input/output) and rhombuses, the subsequent analyses of the proposed methodology.

The geo-statistical Anisotropy needs to be computed on a regular grid and therefore, both wind velocity ($V_w$) and significant wave height ($H_s$) have been interpolated on a rectangular mesh, first on a grid of 1km then to a finer mesh of 10m.

The interpolation method used in this case is the inverse distance weighted (IDW) interpolation, that estimates the value at an interpolated location ($x$) as the weighted average of neighbouring points with weights $w(x)$ given by

$$w(x) = \frac{1}{d(x, x_i)^p}. \tag{3}$$

Here, $x_i$ is a neighbour point, $d$ is the Euclidean distance and $p$ is the inverse distance weighting power. The IDW power chosen is 1 for $R_{V_w}$ and 3 for $R_{H_s}$ and $h$, based on a sensitivity analysis for this area and consistent with the physical relation between wind velocity and generated wave height.

Heatmaps are used to represent the spatial distribution of the geo-statistical anisotropy, showing how the density of $R$ behaves

10 as a function of distance to the coast and time (see Figs. 6 and 7). These maps are scatter plots that act as a 2D-histogram, in which two variables (in this case, $R$ and distance to the coast) are grouped in pre-defined intervals. The elements selected to aggregate samples for the heatmap are hexagons with side 5km and a scale for anisotropy of 20 units for both $R_{V_w}$ and

$R_{H_s}$. Both $R$ and its variance are calculated on a discrete number of distances to the coastline, assuming that the width of the fringe affected by boundary effects is below 100km for this coastal sector (Sánchez-Arcilla and Simpson, 2002). From here, as is with significant wave height to determine the presence of wave-storms (Eastoe et al., 2013; Bernardara et al., 2014), the proposed coastal zone limit is the cutting point where the variance of $R$ is equal to the $90^{th}$ percentile of the total $R$ variance spanning a fringe between 0 and 100km:

$$l = 90^{th} \; percentile \; of \; \text{var}\left(R_{H_s}\left(0\text{km} \leq x \leq 100\text{km}\right)\right) \tag{4}$$

This cutting point has shown, as expected, larger stability for the wave field than for the forcing wind patterns. The variation of $R_{H_s}$ with coastal distance $x$ (Fig. 7) indicates for reference the 20km distance where satellite data offer enough robustness (Cavaleri and Sclavo, 2006; Janssen et al., 2007; Durrant et al., 2009). The plot also displays depth against $x$. The obtained $R_{V_w}$ and $R_{H_s}$ values have been fit to a probability distribution, selecting empirically the lognormal function for the inverse of $R_{V_w}$ and the log-logistic function for $R_{H_s}$. Once estimated the marginal distributions, the dependence structure of the joint probability is adjusted to a Gaussian copula (see Sec. 2).

## 5   Results

The modelled wave heights ($H_s$) have been validated with buoys from the Puertos del Estado monitoring network and available altimeter data (Jason-2, Cryosat and Jason-3). Two locations have been selected, located at the southern (Tarragona) and northern (Begur) coastal sectors (Figs. 4 and 5). The $H_s$ buoy data show good agreement with the simulated $H_s$, quantified in table 1 and in Figs. 4 and 5.

In general, the wave model performs better at deep waters than in coastal waters. The standard deviation is higher in the model than in the observations. At Begur, the bias and the Scatter Index are lower, whereas the RMSE is higher (Tab. 1). At the same buoy, the correlation coefficient is near 95% and the difference between standard deviations is lower (0.2m vs. 0.4m). Note that the Northern part of the Catalan coast is more energetic than the Southern one (see Fig. 5). For instance, in Begur the storm peaks can reach about 7m, whereas at Tarragona the highest recordings are 3.5m.

The altimeter collocated has a positive bias in the coastal zone, and the opposite (i.e. negative bias) happens at deep waters. Nevertheless, there exists qualitative consistency between the in-situ and remote-sensing sources. Additionally, SWAN has been able to capture the regime switching and the proper timing of the storms, despite it tends to underestimate the magnitude of the storm peaks.

$R_{V_w}$ and $R_{H_s}$ have been analysed with heatmaps (Figs. 6 and 7) and scatterplots (Figs. 8 and 9). $R_{V_w}$ presents values that span from 1 to 250 and display a dependence on coastal distance (Fig. 6), featuring a combination of anisotropy close to the land boundary (0 to 20km) and then more isotropic behaviour towards the offshore (up to 100km), although with a rich variability. The wind fields present, in summary, a decreasing variance from 0 to 100km with a pronounced slope from 0 to about 40km (southern sector) or even further offshore (northern sector) and then an almost asymptotical trend. $R_{H_s}$ behaves

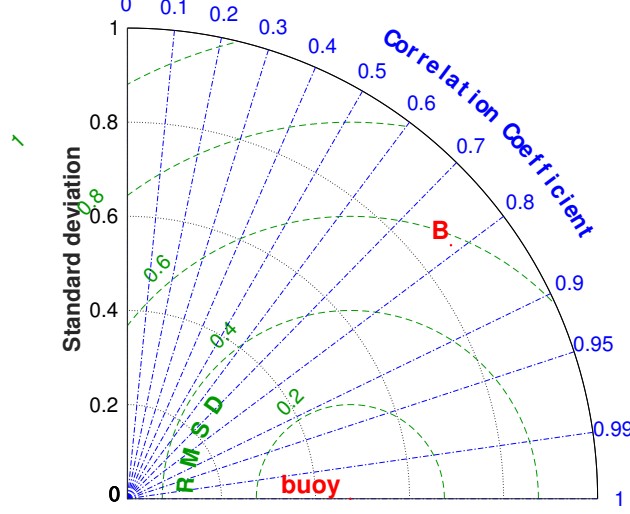

[a]

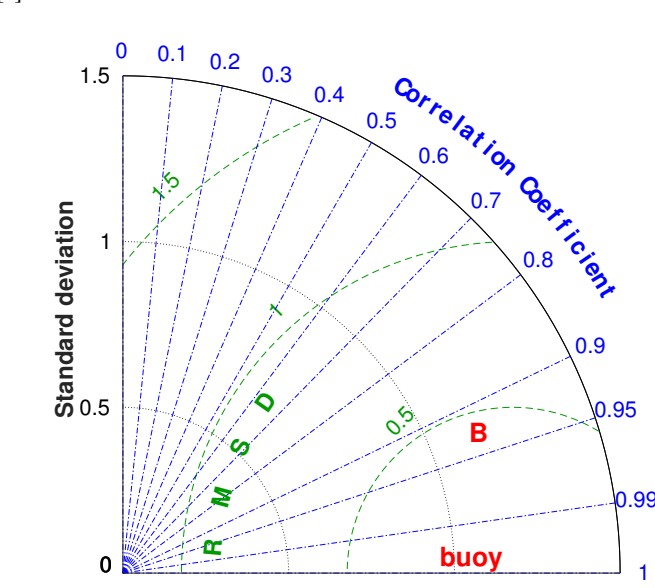

[b]

**Figure 4.** Taylor diagram for the significant wave height ($H_s$) showing correlation, standard deviation and root mean square error (R.M.S.E.) between numerical and observed data for a) south sector (Tarragona location) and b) north sector (Begur location). The time period ranges from November of 2016 to March of 2017.

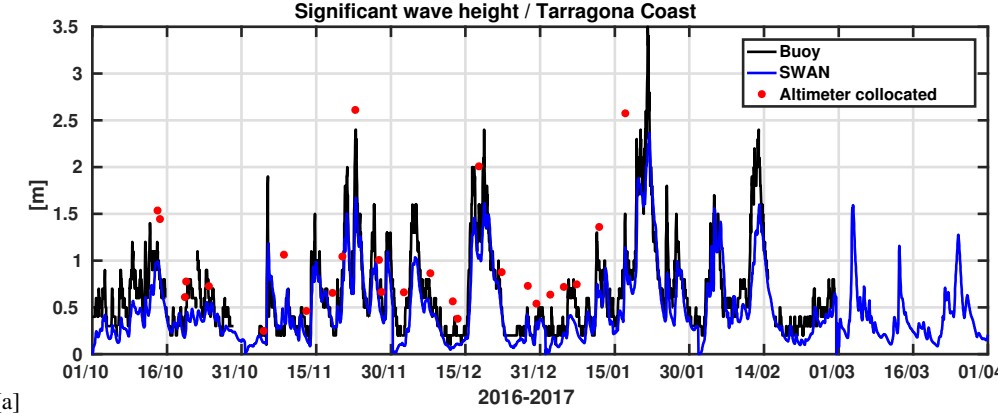

[a]

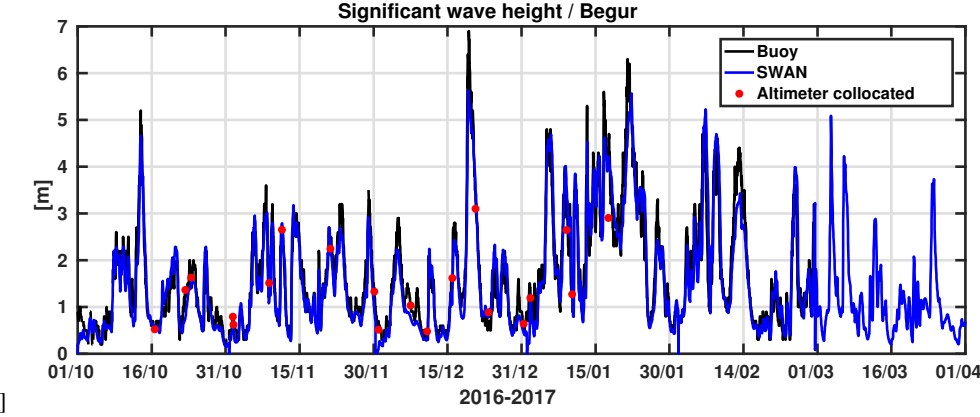

[b]

**Figure 5.** Comparison of numerically simulated significant wave height (SWAN model) with observations, for a) south sector (Tarragona location) and b) north sector (Begur location), for the period October of 2016 to March of 2017. The red dots are altimeter data from the available altimeter data (Jason-2, Jason-3 and Cryosat).

**Table 1.** Statistics of the agreement between numerical significant wave height fields (SWAN model) and observations in terms of root mean square error (R.M.S.E.), bias and scatter index (S.I.) for the control points at the southern and northern coastal sectors.

| Buoy | R.M.S.E. [m] | bias [m] | S.I. [%] |
|---|---|---|---|
| Tarragona coastal buoy | 0.248 | −0.132 | 0.502 |
| Begur | 0.393 | −0.087 | 0.249 |

similarly to $R_{V_w}$, but with a turning point at about 40km in all transects (Figs. 8 and 9) and, thus, a higher level of consistency. From December to January, there are some winds and waves registered within 20km of the coast that present higher $R_{V_w}$ and

$R_{H_s}$, however, they are so few that the variances of $R_{V_w}$ and $R_{H_s}$ in this area do not differ from warmer seasons. Therefore, there is not a clear seasonality to the $R_{V_w}$ or the $R_{H_s}$. The coastal zone limit $l$, corresponding to the $90^{th}$ percentile of the total variance (fringe between 0 and 100km), is calculated from equation 4 (Figs. 8 and 9) and is 3km. It is consistent with time interval (month of study) and location (sector).

In order to find a copula structure, marginal probability distributions for the two anisotropies are needed. Skewness and kurtosis from the analysed data show that the inverse anisotropy of $V_w$ follows a log-normal distribution, while the anisotropy of $H_s$ follows a log-logit distribution. Quantile-quantile plots have been used to assess the fit of each probability function (not shown here) to its target dataset, verifying that the selected samples can be adjusted to the corresponding probability distributions. The joint probability structure of the two anisotropies does not present any marked dependence for the upper-tail

percentiles, suggesting the use of a Gaussian copula, whose dependence parameter $\rho$ is shown in Fig. 10. The so obtained dependence ranges from total independence (0) to a mild ($|\rho_{12}| = 0.1$) dependence between $R_{V_w}$ and $R_{H_s}$.

## 6    Discussion

The calculated anisotropies should be as robust as the starting wave or wind fields that are employed in the analysis. Because of that, the SWAN code has been calibrated with local atmospheric and hydrodynamic conditions (Pallarés et al., 2017).

Special emphasis has been put on using high quality wind fields, both for the direct assessment linked to meteo fields and for the indirect effect they exert on the behaviour of the forced hydrodynamics. The results show, as expected, a higher level of robustness for the wave-based geo-statistical anisotropy, where the calculations used an unstructured grid and a locally adjusted whitecapping term calibration (Pallarés et al., 2014). The cell size has been determined as a function of depth and distance to the coast, consistently with the transect analyses performed in the paper. The application of an unstructured grid

allows reducing computational costs (by about 50%) and the troublesome imposition of internal boundaries.

This leads to an efficient determination of the coastal water boundary that contains some of its common geometric settings (e.g. bathymetric gradients affecting wave fields). Other processes, such as for instance the continental discharge, are of course not captured by the present analysis and would require a similar approach based on the resulting circulation fields, which would certainly capture the regions of fresh-water influence and wave-current interactions (Staneva et al., 2016). However,

the performance of the wave model has shown commonalities with previous studies. For instance, the good performance of spectral models at the Begur buoy can be found in multi-model comparisons (see Bertotti et al. (2012)); and so the consistent underestimation under storm peaks (Cavaleri, 2009).

The anisotropy-based approach will lead to different results depending on met-ocean conditions (wave conditions in our case), requiring a reliable simulation of both average and extreme patterns, as shown by the validation process (see e.g. Figs. 4

and 5). The transfer of energy from the coastal to the offshore domain and vice-versa may condition the results of the analysis for areas near the transition, which is where the boundary will be likely located. This suggests a combined approach, using numerical fields and satellite data supplemented by along-track in situ observations, all suitably interpolated in space and time to provide a picture that is as consistent as possible.

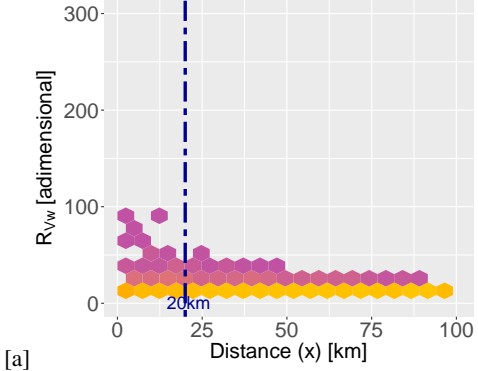

[a]

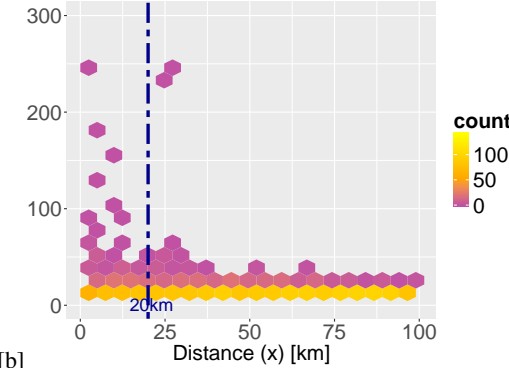

[b]

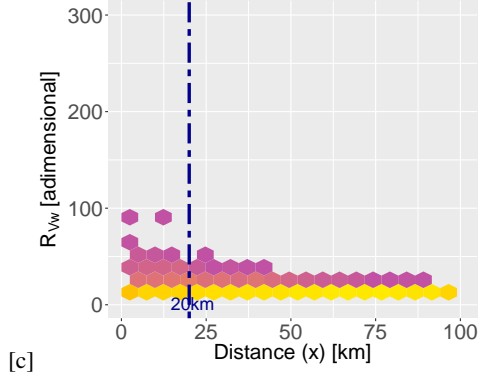

[c]

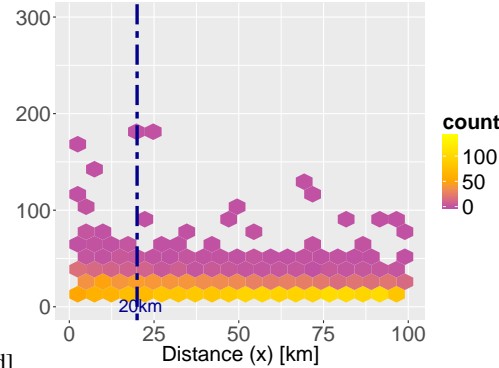

[d]

**Figure 6.** Heatmap of the geo-statistical anisotropy ratio of the wind velocity ($R_{V_w}$) against distance to the coast for a) south control transect (near the Ebre delta), b) central-south transect (near Tarragona harbour), c) central-north transect (near Mataro harbour) and d) north control transect (near Begur cape). The elements selected to aggregate samples for the heatmap are hexagons with side 5km and a scale for anisotropy of 20 units. The counts are the number of elements within a hexagon. A limit of rough order of magnitude for the direct applicability of remote-sensing data (20km) is also shown (blue dashed line). All plots correspond to February 2017.

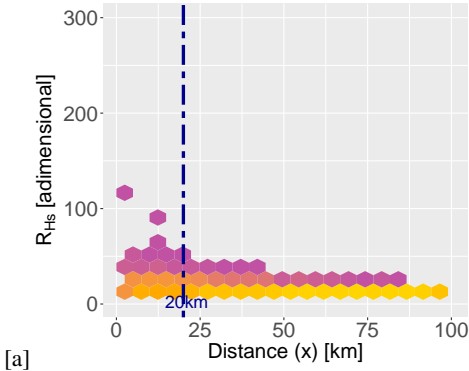

[a]

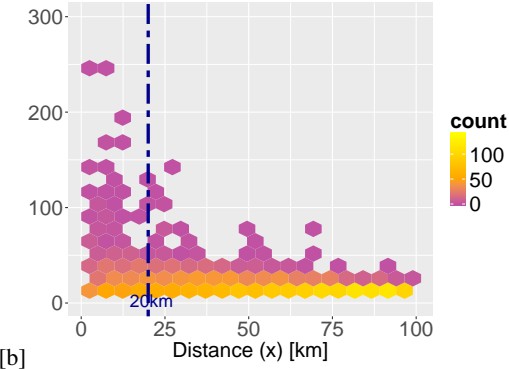

[b]

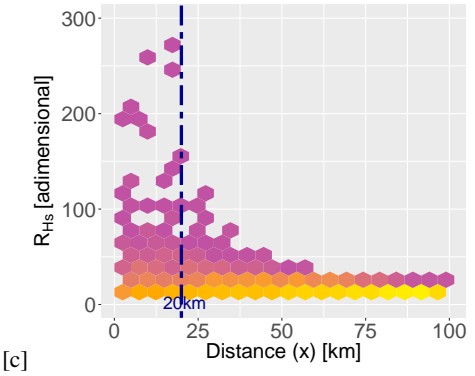

[c]

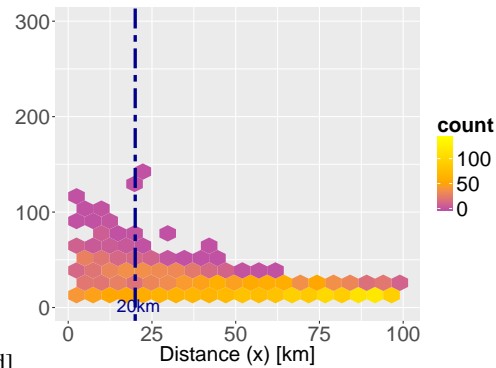

[d]

**Figure 7.** Heatmap of the geo-statistical anisotropy ratio of significant wave height ($R_{H_s}$) against distance to the coast for a) south control transect (near the Ebre delta), b) central-south transect (near Tarragona harbour), c) central-north transect (near Mataro harbour) and d) north control transect (near Begur cape). The elements selected to aggregate samples for the heatmap are hexagons with side 5km and a scale for anisotropy of 20 units. The counts are the number of elements within a hexagon. A limit of rough order of magnitude for the direct applicability of remote-sensing data (20km) is also shown (blue dashed line). All plots correspond to February 2017.

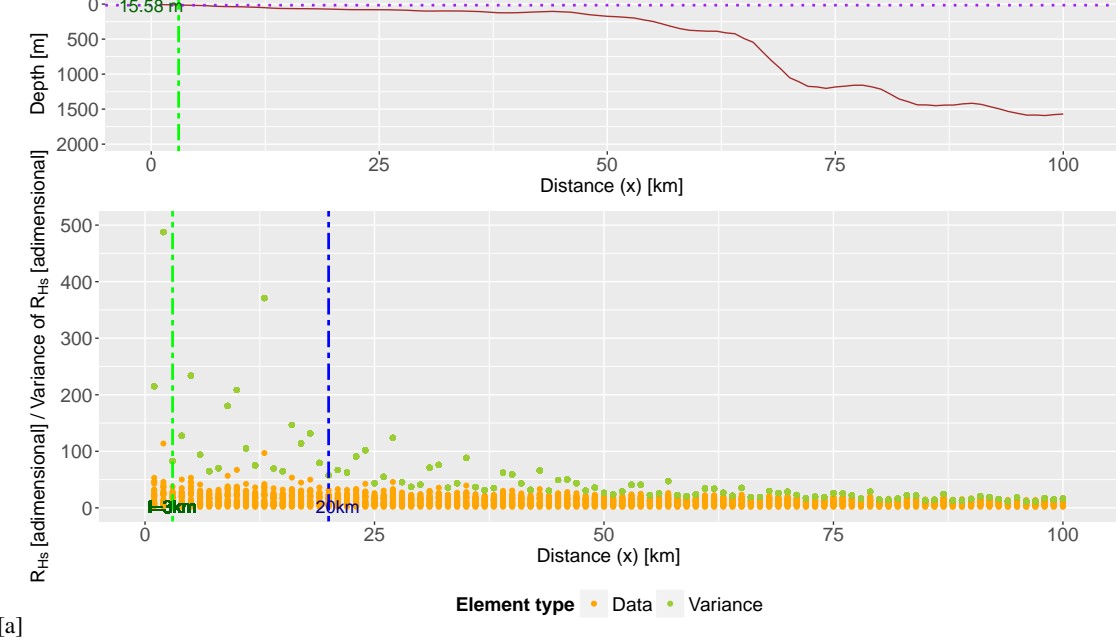

[a]

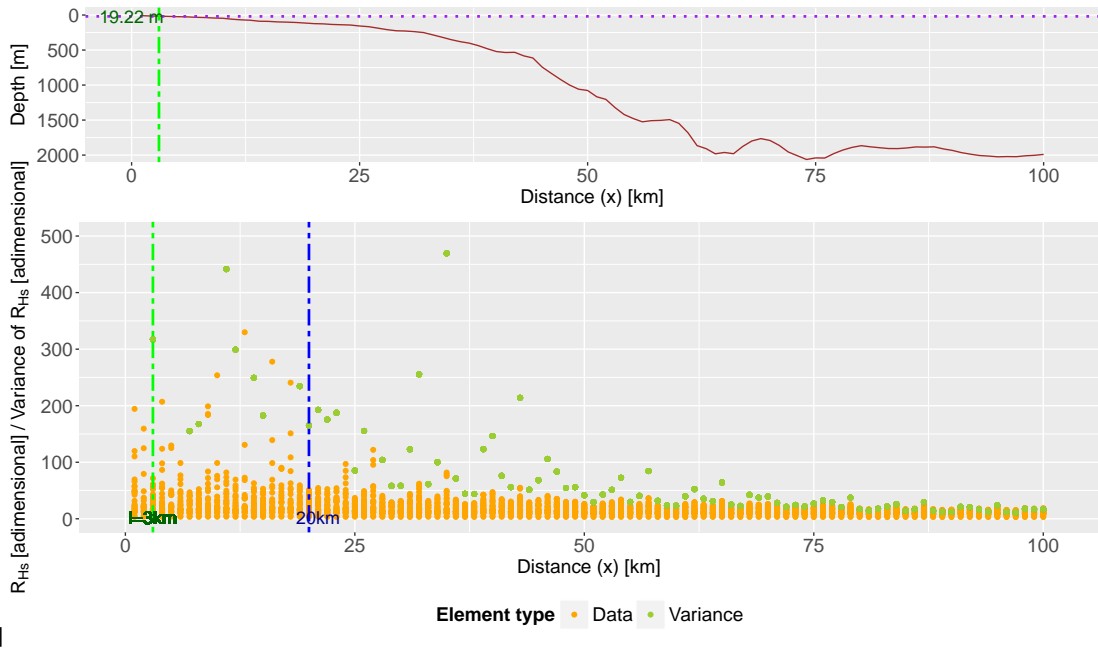

[b]

**Figure 8.** Relation for winter conditions (February 2017) between distance to the coast $x$, depth (upper plot) and anisotropy ratio of significant wave height ($R_{H_s}$), from shore to 100km offshore. Locations are a) south control transect (near the Ebre delta) and b) central-north transect (near Mataro harbour). The distance of 20km which has been suggested as a rough order of magnitude limit for direct applicability of remote-sensing data is also shown (blue dashed line) together with the variance of $R_{H_s}$ across the transect. From here, the coastal zone anisotropy-based boundary has been calculated and is also depicted. A green dash-dot line delimits its horizontal distance from the coast, whereas a purple dotted line denotes its elevation.

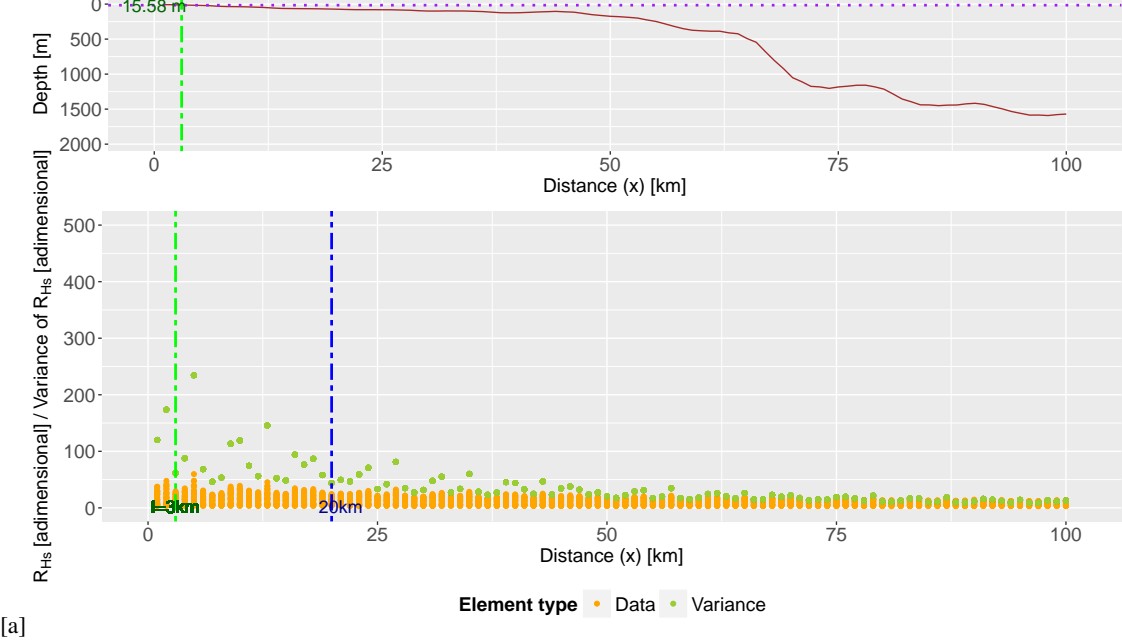

[a]

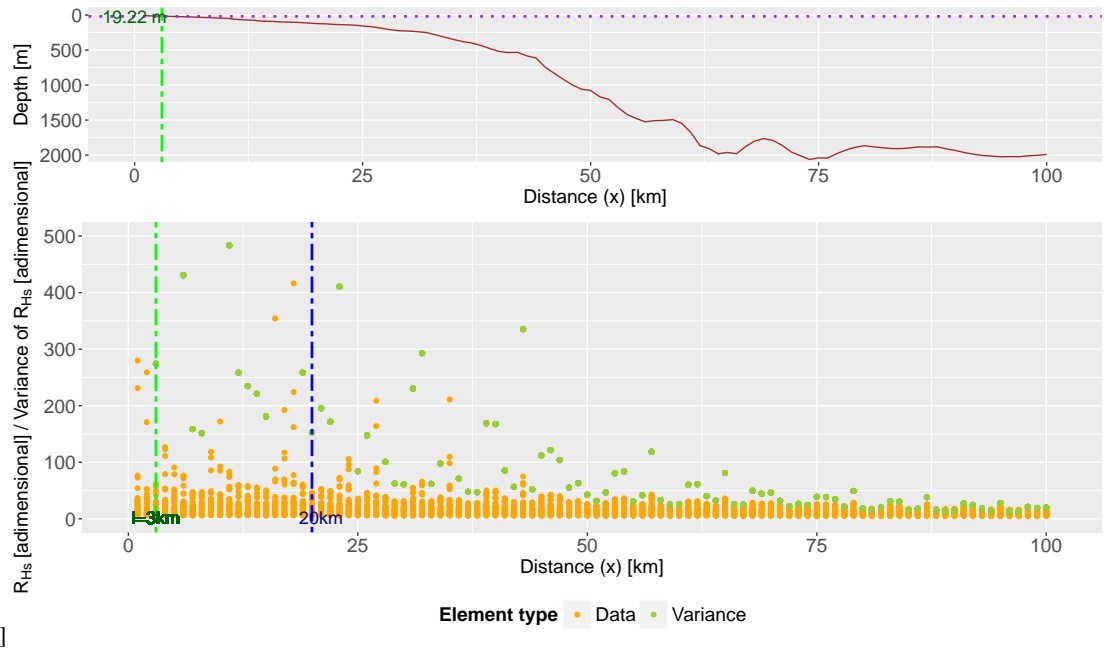

[b]

**Figure 9.** Relation for autumn conditions (November 2016) between distance to the coast $x$, depth (upper plot) and geo-statistical anisotropy ratio of significant wave height ($R_{H_s}$), from shore to 100km offshore. Locations are a) south control transect (near the Ebre delta) and b) central-north transect (near Mataro harbour). The distance of 20km which has been suggested as a rough order of magnitude limit for direct applicability of remote-sensing data is also shown (blue dashed line) together with the variance of $R_{H_s}$ across the transect. From here, the coastal zone anisotropy-based boundary has been calculated and is also depicted. A green dash-dot line delimits its horizontal distance from the coast, whereas a purple dotted line denotes its elevation.

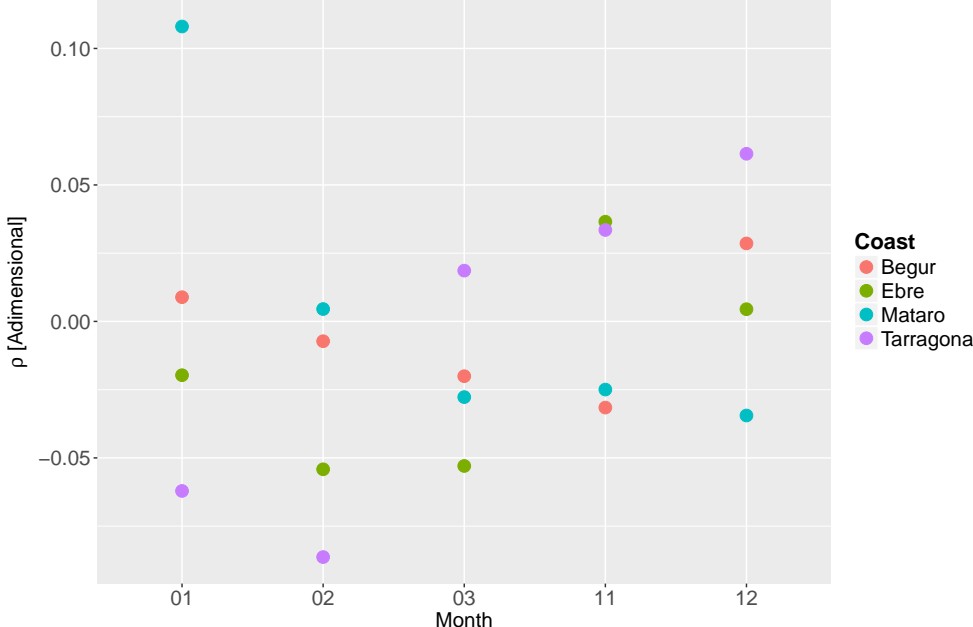

**Figure 10.** Copula parameters $\rho$ of the proposed Gaussian copulas for all considered profiles: a) south control transect (near the Ebre delta), b) central-south transect (near Tarragona harbour), c) central-north transect (near Mataro harbour) and d) north control transect (near Begur cape). The plot shows the variation with time (horizontal axis) between November of 2016 and March of 2017. The parameters are placed in a manner that they start from January.

The values of $V_w$, $H_s$ and $h$ obtained through the IDW interpolation, using an IDW weighting from 1 to 3, are similar and reasonable. For marine variables the weight of 3 has been selected, to account for the influence of the closest neighbours based on the water inertia (which is 3 orders of magnitude larger than for air). The proposed IDW power for $V_w$ is smaller because gas is more turbulent than water and thus should have a smaller spatial dependence. The obtained pattern for $R_{V_w}$ is consistent
for the four selected transects in the study area, showing a mostly isotropic behaviour for coastal distances from 0 to 100km. Higher values of $R_{V_w}$ at distances from the coast below 20km (see Fig. 6) indicate a clear directional spread of winds within the coastal fringe, linked to orographic control such as channelling by local mountains and river valleys.

Despite that the mainly shown results are based on November 2016 and February 2017, the spatial trends of the anisotropy were coherent throughout the simulation period, thus exhibiting the robustness of the methodology.
The numerical wind fields present errors below $2\mathrm{m/s}$ (Martin et al., 2006), which means that the $V_w$ calculated can represent well actual wind conditions. The $R_{V_w}$ is lower in Ebro Delta (Fig. 6 [a]) than at Tarragona (Fig. 6 [b]) due to the following reason: the orography at the Ebro delta is flat, and wind blows from a wide range of directions; whereas Tarragona features mountain chains that channel the winds into a more limited directional subset. The anisotropy pattern at Begur (Fig. 6 [d]) may be explained due to the strong Mistral and Eastern winds (Obermann et al., 2016; Obermann-Hellhund et al., 2017), that both
affect nearshore and deep waters.

Hence, the behaviour of $R_{V_w}$ near the coast can show sharp local variations due to the joint effect of orography, mesoscale circulation and large-scale circulation, all affecting wind strength and directionality. However, the seasonality does not affect $R_{V_w}$ as much. In all cases $V_w$ becomes more isotropic towards the offshore, denoting a decreasing control by the land-water boundary.

The $R_{H_s}$ pattern is similar, with wave fields showing boundary effects (mainly in directional properties) for coastal distances below 50km, which has also been considered an order of magnitude estimate for land wind effects. Farther from the coast there is a clear trend to isotropy, more pronounced for transects with more stable atmospheric conditions. The link between $R_{H_s}$ and land-originated winds can be appreciated by the shift from homogeneous to anisotropic behaviour in the northern most transect (Begur) where the effect of $V_w$ on $H_s$ is only evident up to 38km from the coast. Note that in the Ebro Delta shows a more isotropic value at the coastal zone. This area presents a wider continental shelf than the other profiles, then wave dissipation and refraction tend to be more homogeneous. Henceforth, more anisotropic winds plus a steeper profile may be considered as the main reasons for the discrepancies among the beach profiles.

Such behaviour of $R_{H_s}$ with coastal distance parallels that of particulate matter diffusivity, which tends to become isotropic at around 10km (Romero et al., 2013) from the land-water boundary. The degree of geo-statistical anisotropy in diffusivity is physically related to eddy kinetic energy, which varies as $x^{5/4}$ ($x$ being separation distance) for depths below 20m (Gràcia et al., 1999). A steep bottom slope will favour deep-water wave behaviour at relatively short distances from the coast, as shown by the distinctive behaviour of $R_{H_s}$ for coasts of different slope. However, although $R_{H_s}$ presents greater variance (more outliers) for steep slopes (e.g. the due to the combination of deep and shallow water wave regimes as in the central-north transect, near Mataro harbour) the gradient of $R_{H_s}$ with distance to the coast is similar for all types of bathymetry considered, suggesting a generic value of the proposed approach. This is also true for any time of the year.

The coastal boundaries suggested by Sánchez-Arcilla and Simpson (2002) for the Catalan Coast can be 0.1-0.6km (frictional coupling of fluids between shelf and nearshore), 10km (non-linear coupling between shelf and slope), 1km (non-linear coupling between shelf and nearshore), among other suggested values of the same order of magnitude. The $l$ provided in our analysis is slightly larger than the value given for the frictional coupling of fluids between shelf and nearshore, whereas it is similar or smaller than in the non-linear couplings. Nevertheless, the orders of magnitude are similar.

The $\rho$ parameter of the Gaussian copula, characterizing the dependence structure among $R_{V_w}$ and $R_{H_s}$ reflects a certain similarity of the spatial behaviour for both variables, $R_{V_w}$ and $R_{H_s}$ (see Figs. 6 and 7). The overall mutual dependence of $R_{V_w}$ and $R_{H_s}$ is strongest for the northern-most transect (Begur), where the topo-bathymetric control of the Pyrenees and their submerged signature becomes better defined. Such mutual dependence gets weaker for the central and southern coastal transects (see Fig. 10). There seems to be a strong wind channelling aligned with the main river valleys (Sánchez-Arcilla et al., 2014; Rafols et al., 2017) in February and March, dominating the local wave fields. The $\rho$ parameter should reflect this spatial and temporal variation, resulting in a coastal zone width that will be a function of the prevailing met-ocean drivers and should thus be considered as a dynamic concept.

The resulting coastal definition can be data (numerical or observed) driven, being directly applicable to any region with a forecasting system or with enough coverage of in-situ plus satellite data. The proposed criteria appear to work well for wave-

dominated and micro-tidal environments, and although suitable for any combination of factors, their application to macro-tidal regimes or river-discharge dominated areas should account for the corresponding signature in the hydrodynamic fields. Under these conditions the preferred variable could change to current velocity or to temperature, considering in all cases the effect of spatial resolution in the results.

## 7 Conclusions

The proposed coastal fringe (water sub-domain) definition is based on an objective estimation of the geo-statistical anisotropy as a proxy for the influence of the land border. The suggested statistical assessment can be applied to any variable that reflects such an influence (here it has been illustrated with wind velocity and significant wave height) and can be easily automated for any field, numerical or observational, that presents enough resolution.

The methodology has been tested with numerically generated fields, validated with datasets from Puertos del Estado buoys and altimeter. Anisotropies of wind velocity and significant wave heights have been extracted along a set of characteristic profiles spanning widths up to $100$km (see Fig. 2), considered sufficient for the relatively narrow shelfs in the Spanish Mediterranean coast. The performed analysis has shown how wind and wave fields are influenced by the land-sea border, demonstrating the topo-bathymetric control on met-ocean factors. This control depends on topographic (mountain chains and river valleys) and bathymetric (bottom slope, submarine canyons or prodeltas) features but also on the energetic level of the prevailing weather, leading to a dynamic definition of the coastal water domain. The resulting widths, based on variance variation, span distances in the kilometre range, depending on bottom slope and coastal plan shape geometry. The correlation between the wind and wave based definitions (i.e. the mutual dependence among $R_{V_w}$ and $R_{H_s}$) seems to be stronger in the northern-most parts of the study area, where the topo-bathymetric control is most prominent.

This new definition of the coastal zone can be useful for setting up standards to delimitate this transitional fringe, facilitating the selection of processes and boundary conditions for modelling and providing an objective coastal zone limit for impact assessments. Such an approach can also support directional and asymmetric measures of error and the underlying metrics (between model and data), leading to improved products and standards in the coastal zone.

*Acknowledgements.* This paper has been supported by the European project CEASELESS (H2020-730030-CEASELESS) and the Spanish national projects COBALTO (CTM2017-88036-R) and ECOSISTEMA-BC (CTM2017-84275-R). As a group, we would like to thank the Secretary of Universities and Research of the department of Economics of the Catalan Generalitat (Ref. 2014SGR1253). We duly thank the Meteorological Office of the United Kingdom for the provided wind fields, Copernicus Marine Environment Monitoring Service for the altimeter data, and Ente Publico Puertos del Estado for the in-situ measurements.

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
