# Peer review of "The land-sea coastal border: A quantitative definition by considering the wind and wave conditions in a wave-dominated, micro-tidal environment"

_Ocean Science, 2018_

## Referee Comment (RC1) · Anonymous Referee #1 · 5 Oct 2018

GENERAL COMMENTS This study is aimed at assessing the definition (say distance from the coast) of land-sea boundary. For this purpose, wind (from UKMO) and wave (using SWAN) model data are used. No additional oceanic/atmospheric variables are taken into account. The whole analysis is based on the calculation of anisotropy of 2-D fields, which is computed along four transects of the Catalan Coast (used as test site). The paper is well-written, but it lacks a connection with other methods currently employed to define that boundary (e.g. using oceanic variables, as salinity, or depth) and uses an unique strategy (anisotropy + quantile threshold) for the identification of the boundary. In my opinion the paper can be improved and worthy to be published after a major revision that assesses the stability of the boundary computation and its seasonality.

[Figure]

SPECIFIC COMMENTS Title. Since the definition is based on wind and waves I suggest mentioning them in the title. Or at least making it clear that in the study a methodology is proposed (which can be then used with other variables).

Abstract. o "The more robust estimator [. . .]". How the robustness of the estimator is expressed? Did authors check, for example, the sensitivity of the results on the quantile threshold? o Can authors show the distribution function of anisotropy for wave and wind fields?

Introduction. o The land-sea border problem is presented by means of references mostly pointing to the same group that wrote the paper. My suggestion is to improve the overall view of the problem. o It seems to me that other references are more appropriate for the SWAN model.

Theorethical background. o I warmly suggest improving this section to make the reader more familiar with the concept of anisotropy. A couple of examples (high anisotropy, low anisotropy) using different set of wave/wind data will help to familiarize with the concepts here presented. o The definition of R seems not to be consistent with the one given by Chorti and Hristopulos (2008). I suggest improving the presentation of the method. o Is there a difference between the adopted method and others used to compute the consistency of spatial fields, such as the structure tensor? o For the Copula function, I suggest changing $(u, v)$-variables with other symbols as they are already used for the definition of anisotropy.

Methods o How the covariance of anisotropy is computed? o I suspect the wind model resolution is not $2.5°$ and $3.75°$. However, were they km (instead of degrees), the model seems not to be enough resolved to provide accurate data at 2-3 km scale, which is the final value provided for the border. The SWAN model at 600 m close the shoreline is at limit in this respect (the border encompasses 3-4 grid points). Can authors comment on this aspect? o If you define the resolution in meters, I guess it is smaller near the coast, not higher. o Which period is spanned by the analysis? (February 2017? Why

not using a longer period, say 2016-2017?). o As far as waves are concerned, using the proposed methodology, the distance of the border should change between August (instead of November, Fig. 9) and February. A season-based classification of the border would be a sound improvement of the paper. o About the radius and quantile threshold. Those two values seem to me quite arbitrary and not directly physically-based. Authors could do a sensitivity analysis to show how the results depend upon those values (the quintile, in particular). This is an important task in order to evaluate the stability of the proposed methodology (and then make it usable in other contexts) and provide the uncertainty of the location of the boundary. o It would be useful to plot the quantile values on the heatmaps. o Please explain what is the count in the heatmap's caption? o I suggest putting the wave model assessment in a dedicated sub-section separated from the results of the anisotropy analysis. o Is there a reason why in the panels of Fig 8 and 9 the dashed green vertical lines are not aligned (horizontal axes seem consistent)?

---

## Referee Comment (RC2) · Anonymous Referee #2 · 5 Nov 2018

Review: "The land-sea coastal border: A quantitative definition", by Sánchez-Arcilla et al.

Recommendation: major revision.

Summary: The authors attempt to provide a quantitative and generalizable definition of "land-sea" zone, i.e., cross-shore width of that particular marine area that is strongly affected by the presence of the continent. The methodology is based on the measure of anisotropy of specific, vectorial and/or scalar fields of environmental parameters. For this work, the authors use wind velocity and significant wave height from well-calibrated and validated numerical outputs.

General comments The work the authors present is really intriguing and I particularly

like the idea of defining a "coastal zone" by using environmental variables in a quantitative fashion. However, while the specific variables the authors consider in this application (i.e. wind velocity and significant wave height) are particularly suitable for the study area they might not work for a different environment, where, for instance, wind and wave patterns do not actually characterize the coastal zone. As stated by the authors, river plumes or, more in general, bio-geochemical processes may lead for a better definition of a "coastal area" and, as a consequence, the methodology here proposed might not be suitable. All this is at the base of my main criticism: to state that such a methodology provides a "quantitative definition for the land-sea (coastal) transitional area" is too strong; although I like generalizations, I still believe that a "land-sea (coastal) transitional area" can be defined by starting from the specific physical, and/or biogeochemical, and/or geological, and/or ecological process we want to investigate. A second comment regards the poor connection between the pure mathematical/statistical part and the environmental application. I would have appreciated a better explanation of the statistics by starting from the environmental data, also discussing physical meanings and assumptions. To present the theoretical background as it is leaves the reader with some doubts regarding the feasibility of the methodology.

Specific, minor comments

Abstract - replace "perpendicular" with "cross-shore" in line 2

Introduction - The are several definition of what a Land-sea border is (Shaw et al., 2008; Geleynse et al., 2012). I would avoid (at least, at the beginning) to frame the land-sea border within this specific definition. Instead, it would be better to state that over land-sea border areas occur specific met-ocean dynamics that actually characterize land-sea coastal border. The aim of this work is to quantitative define the extension of this area. (see general comment).

- "Sentinel data" (in line 3-pag 2) ; the general reader might not be familiar with the sentinel missions and, therefore, might not understand that here authors are referring
to satellite data. Please, introduce the Remote Sensing approach properly.

- "Because of that" (in line 5-pag 2); Please, be more specific. It's not clear the use of Sentinel data in defining land-sea limits and what the authors mean with degradation of data. "necessary"; too strong, I would write "useful" rather than necessary.

- "coastal anisotropy" (in line 13-pag 13); I would write "anisotropy of environmental parameters" rather than coastal anisotropy

Theoretical background - G(x) in line 10 should be G(y), as far as I am missing something; As I suggest in the General Comments, this section would be much clearer (and the ms much stronger) if the theoretical back ground is explained by starting from environmental variables. As it is, the reader might get confused.

Study area - By reading the section it comes natural to think that the analysis is particularly suitable for this study area, thus difficult to generalize

References Shaw, J. B., Wolinsky, M. A., Paola, C., & Voller, V. R. (2008). An image‐based method for shoreline mapping on complex coasts. Geophysical Research Letters, 35(12).

Geleynse, N., Voller, V. R., Paola, C., & Ganti, V. (2012). Characterization of river delta shorelines. Geophysical Research Letters, 39(17).

---

## Referee Comment (RC3) · Anonymous Referee #3 · 8 Nov 2018

GENERAL COMMENTS:

The paper addresses a relevant scientific question which is well within the scope of Ocean Science. The authors present a methodology with the aim to quantitatively define the land-sea boundary in wave-dominated and micro-tidal environments. The presented methodology builds on met-ocean datasets which are well-known and frequently used in the field. The authors conclude that the proposed land-sea boundary (coastal fringe) definition is a generic method. However, as also stated by the author, the correct choice of met-ocean or biogeochemical variables might be case dependent and the presented application is tailored to the case specific conditions at the Catalan coast. It would significantly improve the general applicability of the method if the authors could briefly describe how one should choose the variables that reflect the in-

fluence of the land border in a specific application. Moreover, by comparing the results to other land-sea border definitions (validation) and by providing uncertainty estimate of the computed coastal zone limit the authors would make the methodology stronger.

The scientific methods and assumptions are in general valid and clearly outlined, even though further clarifications are required at certain sections (see specific comments). The paper is well structured in general; however, certain elements should be better explained (see specific comments).

SPECIFIC COMMENTS:

Title:

- The title should indicate that the methodology to quantitatively define the land-sea coastal boarder was only tested for a case study in a wave-dominated and micro-tidal environment.

Abstract:

- I propose to change the term "90th quantile" to "90th percentile" throughout the paper. The authors refer to the 90th 100-quantile which is called percentile.

Study area:

- The authors state (page 4, line 5) that the focus area is the Spanish north-eastern Mediterranean coast due to the availability of in-situ and Sentinel images for support. It is not clear how the Sentinel images were utilised in the methodology as a support (unless they were used for the SWAN model validation, which is not stated in the paper).

Methods:

- Further background information on the Unified Model and/or the wind field data should be given. The wave data is explained in much more detail compared to the wind data.

- According to Cullen (1993) the operational forecast grid for the Unified Model is 0,833 degree (latitude) and 1,25 degree (longitude), whereas the standard climate and upper atmosphere configuration uses 2,5 degree (latitude) and 3,75 degree (longitude). Is it really the second configuration which is used in this paper? This resolution would mean approximately 250km (latitude) and 310 km (longitude). That is a very coarse resolution for this purpose.

- I suggest adding steps to the methodology figure (Figure 3) for the wave and wind model validation, interpolation, as well as for the distribution fitting (Gaussian copula model).

- It is mentioned (page 6, line 9) that wave fields have been validated. Validation results should also be included for the wind field data. Reference to the wind field validation is only given in the discussion section (page 15, line 3-4). I suggest moving this sentence to the Methods section where the United Model is described.

- Is there any reason why the 90th percentile is used in equation 6? Is it based on expert knowledge or literature?

Results:

- The figures 6,7,8,9 are presented on pages 11-to 14 while described on page 7. This makes it hard for the reader to follow the paper. Consider to move them closer to the place where they are described.

- In Figure 5 red dots are labelled as Altimeter data. Is this data coming from Sentinel images? If yes, please explain both in the legend and in the text, and also add which mission it is (e.g. 3A).

- The calculated coastal zone limit values are not mentioned explicitly in the results section, even though they are depicted in Figure 8-9 and mentioned in the abstract. I suggest mentioning them in the text as this is the main objective of the methodology.

- Please use the word "coastal zone limit" consistently. Sometimes it is only called

"limit".

Discussion:

- The authors write that "The calculated anisotropies should be as robust as the starting wave or wind fields that are employed in the analysis" – that is why the robustness of the wind field should be better defined in the Methods section.

- Figure 6-7: the description of the hexagons in the heatmap should be added to the figure as they are only described in the text. Also, the description of the blue dashed line at 20 km should be added as in Figure 8 and 9.

- Figure 10: The x axis represents the months within a year cycle. Which year is it? And why only months 1, 2, 3, 11,12 were selected?

- The authors write (page 16, line 5) that the correlation between R_Vw and R_Hs is the strongest for the Begur transect. On the other hand, in Figure 10 the Begur transect (orange dots) has a correlation parameter around 0 (max ∼0.026). This figure indicates that the Mataro transect has the strongest correlation parameter, not Begur. I suggest clarifying this.

References:

- The number of references is rather high (57). Moreover the share of references originating from the same authors is also high.

---

## Short Comment (SC1) · 8 Nov 2018

The authors present a methodology for determining the land-sea transitional area based on the empirical distribution of anisotropy in meteorological and ocean processes. This is an interesting article, however it will be beneficial for the audience if the authors could provide some feedback on the following matters:

**1 Definition of anisotropy**

In [Chorti et.al., 2008] a non-parametric estimator of statistical anisotropy was proposed, for which an approximate estimate of the anisotropy statistics distribution was

provided in [Petrakis et.al., 2017]. While the authors cite [Chorti et.al., 2008], from the rest of the references it is not clear if anisotropy is defined as in geostatistics (statistical anisotropy: directional dependence of correlation functions) or as in (geo)physics (directional variation of a physical property, e.g., elasticity, permittivity). Also it is not clear how anisotropy is estimated. The authors should clarify, by providing the definition of anisotropy and the estimator they use.

**2 Spatial resolution of wind and wave fields**

For both fields there are sub-domains with anisotropy ratio estimates of $R \approx 100$ or more. Therefore, the largest correlation length within such sub-domains is larger by two orders of magnitude compared to the smallest correlation length over the perpendicular principal axis. Assuming stationarity, for an accurate estimation of anisotropy a field should be sampled at a sufficiently large domain, to satisfy ergodicity, and at a high resolution, in order to capture the spatial variability at length scales below the smallest correlation length. The authors estimate anisotropy over circular sub-domains of 5km radius. Some representative field maps would be useful to justify that the sub-domains are sufficiently large and contain an adequate number of measurement samples for the fulfillment of the aforementioned requirements.

**References**

Chorti, A., Hristopulos, D. T. (2008). Nonparametric identification of anisotropic (elliptic) correlations in spatially distributed data sets. IEEE Transactions on Signal Processing, 56(10 I), 4738-4751. doi:10.1109/TSP.2008.924144

Petrakis, M. P., Hristopulos, D. T. (2017). Non-parametric approximations for

anisotropy estimation in two-dimensional differentiable gaussian random fields. Stochastic Environmental Research and Risk Assessment, 31(7), 1853-1870. doi:10.1007/s00477-016-1361-0

---

## Author Comment (AC1) · 9 Jan 2019

**Reviewer #1**

**GENERAL COMMENTS**

*This study is aimed at assessing the definition (say distance from the coast) of land-sea boundary. For this purpose, wind (from UKMO) and wave (using SWAN) model data are used. No additional oceanic/atmospheric variables are taken into account. The whole analysis is based on the calculation of anisotropy of 2-D fields, which is computed along four transects of the Catalan Coast (used as test site). The paper is well-written, but it lacks a connection with other methods currently employed to define that boundary (e.g. using oceanic variables, as salinity, or depth) and uses an unique strategy (anisotropy + quantile threshold) for the identification of the boundary. In my opinion the paper can be improved and worthy to be published after a major revision that assesses the stability of the boundary computation and its seasonality.*

**SPECIFIC COMMENTS**

**Title.**

*Since the definition is based on wind and waves I suggest mentioning them in the title. Or at least making it clear that in the study a methodology is proposed (which can be then used with other variables).*

Thank you so much for the suggestion. We have adapted our title to it. The article has been renamed as: "The land-sea coastal border: A quantitative definition by considering the wind and wave conditions in a wave-dominated, micro-tidal environment."

**Abstract.**

*"The more robust estimator [...]". How the robustness of the estimators expressed? Did authors check, for example, the sensitivity of the results on the quantile threshold?*

(pp. 1, line 6) We have changed "robust" by "viable", as the selection of the 90th percentile is a convention commonly followed in Literature (Eastoe 2013, Bernadara 2013).

*Can authors show the distribution function of anisotropy for wave and wind fields?*

(pp. 1, line 7) It is the 90th quantile of the variance of the anisotropies. We have

rectified by introducing  this specification in the text.

**Introduction.**

**The land-sea border problem is presented by means of references mostly pointing to the same group that wrote the paper. My suggestion is to improve the overall view of the problem. o It seems to me that other references are more appropriate for the SWAN model.**

The state-of-the-art has been improved with recents works in the same study area.

The number of citations from the same group has been reduced. Here is a list of the ones that have been obviated:
-Bolaños and Sánchez-Arcilla (2006), as it can be represented by Bolaños et al. (2009).
-Bolaños et al. (2007), as it only appears once in the text and along other references.
-Pallarés et al. (2013), for the same reason.
-The thesis of E. Pallarés can be represented by Pallarés et al. (2014).
-Sánchez-Arcilla et al. (2008), as it is similar to Bolaños et al. (2009).
-Sánchez-Arcilla et al. (2016), as it only appears once, and along with another reference. Also, it is more about ports.
-Sierra et al. (2017) has been obviated, as it is well represented by the other bibliography that accompany it in the "Introduction".

It has been added references, not only about the SWAN model, but on spectral wave modelling as well:

*Bertotti, L., Bidlot, J., Bunney, C., Cavaleri, L., Passeri, L. D., Gomez, M., Lefe, J., Paccagnella, T., Torrisi, L., Valentini, A., and Vocino, A.: Performance of different forecast systems in an exceptional storm in the Western Mediterranean Sea, Quarterly Journal of the Royal Meteorological Society, 138, 34–55, https://doi.org/10.1002/qj.892, https://rmets.onlinelibrary.wiley.com/doi/abs/10.1002/qj.892, 2012.*

*Booij, N., Ris, R., and Holthuijsen, L.: A third-generation wave model for coastal regions, Part I, Model description and validation, Journal of Geophysical Research, 104 (C4), 7649–7666, 1999.*

*Cavaleri, L., Bertotti, L., and Pezzutto, P.: Accuracy of altimeter data in inner and coastal seas, Ocean Science Discussions, 2018, 1–13, https://doi.org/10.5194/os-2018-81, https://www.ocean-sci-discuss.net/os-2018-81/, 2018.*

*Lionello, P. and Sanna, A.: Mediterranean wave climate variability and its links with NAO and Indian Monsoon, Climate Dynamics, 25, 611–623,*

https://doi.org/10.1007/s00382-005-0025-4, 2005.

Qi, J., Chen, C., Beardsley, R. C., Perrie, W., Cowles, G. W., and Lai, Z.: An unstructured-grid finite-volume surface wave model (FVCOM-SWAVE): Implementation, validations and applications, Ocean Modelling, 28, 153 – 166, https://doi.org/https://doi.org/10.1016/j.ocemod.2009.01.007, http://www.sciencedirect.com/science/article/pii/S1463500309000067, the Sixth International Workshop on Unstructured Mesh Numerical Modelling of Coastal, Shelf and Ocean Flows, 2009.

Roland, A. and Ardhuin, F.: On the developments of spectral wave models: Numerics and parameterizations for the coastal ocean, Ocean Dynamics, 64, 833–846, 2014.

Roland, A., Zhang, Y. J., Wang, H. V., Meng, Y., Teng, Y.-C., Maderich, V., Brovchenko, I., Dutour-Sikiric, M., and Zanke, U.: A fully coupled 3D wave-current interaction model on unstructured grids, Journal of Geophysical Research: Oceans, 117, https://doi.org/10.1029/2012JC007952, https://agupubs.onlinelibrary.wiley.com/doi/abs/10.1029/2012JC007952, 2012.

Staneva, J., Wahle, K., Günther, H., and Stanev, E.: Coupling of wave and circulation models in coastal-ocean predicting systems: a case study for the German Bight, Ocean Science, 12, 797–806, https://doi.org/10.5194/os-12-797-2016, https://www.ocean-sci.net/12/797/2016/, 2016.

Wiese, A., Staneva, J., Schulz-Stellenfleth, J., Behrens, A., Fenoglio-Marc, L., and Bidlot, J.-R.: Synergy of wind wave model simulations and satellite observations during extreme events, Ocean Science, 14, 1503–1521, https://doi.org/10.5194/os-14-1503-2018, https://www.ocean-sci.net/14/1503/2018/, 2018.

WISE Group: Wave modelling-the state of the art, Prog. Oceanogr., 75, 603–674, 2007.

Zijlema, M.: Computation of wind-wave spectra in coastal waters with {SWAN} on unstructured grids, Coastal Engineering, 57, 267 – 277, https://doi.org/http://dx.doi.org/10.1016/j.coastaleng.2009.10.011, http://www.sciencedirect.com/science/article/pii/S0378383909001616, 2010.

**_Theoretical background._**

_I warmly suggest improving this section to make the reader more familiar with the concept of anisotropy. A couple of examples (high anisotropy, low anisotropy) using different set of wave/wind data will help to familiarize with the concepts here presented._

A clarification has been added to the introduction (pp. 2, line 15): "A wind or wave field that has a high anisotropy can present a predominant wind or wave direction, respectively." The interpretation of the geometric anisotropy values and the possible underlying physical processes are included throughout the Discussion section.

(pp. 15, lines 5-6) The phrase: " which facilitates multiple simulations to perform a statistically stable analysis of anisotropy" is misleading and is not very much informative. Thus, it is eliminated.

_The definition of R seems not to be consistent with the one given by Chorti and Hristopulos (2008). I suggest improving the presentation of the method._
(pp. 3, lines 2-7) We have improved the definition as: "Given a spatio-temporal field $X(s, t)$, where s stands for a 2-D vector (zonal and meridional components) and t is the time, it is assumed that the iso-level contours of the correlation functions are invariant, i.e. ellipses in two dimensions. The main axis of these ellipses are termed u and v, respectively (see Fig. 1). The metric of the geometric anisotropy, then, becomes their ratio $R = u / v$ (R exists [0, inf)) (Chorti and Hristopulos, 2008; Petrakis and Hristopulos, 2017). An R value close to unity means that u and v are isotropic, i.e. homogeneous across the different directional sectors. As R increases, the difference between the main axis increase, showing higher anisotropy at certain directional sectors."

_Is there a difference between the adopted method and others used to compute the consistency of spatial fields, such as the structure tensor?_
We understand that the structure tensor is a similar concept to the anisotropy, but what we do here is to put emphasis on the anisotropy of the wind and wave conditions near the coast and to statistically quantify its spatial distribution.

_For the Copula function, I suggest changing (u, v)-variables with other symbols as they are already used for the definition of anisotropy._
We have proceeded as suggested.

**Methods**

**How the covariance of anisotropy is computed?**

The covariance matrix is computed following (Chorti and Hristopulos, 2008):

From a X(s,t), where s is a 2D-vector (zonal (i) and meridional (j) component) and t is the time. We assume, for each time step:

  1. The covariance matrix Q can be computed: $Q(i,j) = E\ [\ dX(s)/ds(i);\ dX(s)/ds(j)]$

Where E[·] are the ensemble averages.

*I suspect the wind model resolution is not 2.5 and 3.75 . However, were they km (instead of degrees), the model seems not to be enough resolved to provide accurate data at 2-3 km scale, which is the final value provided for the border. The SWAN model at 600 m close the shoreline is at limit in this respect (the border encompasses 3-4 grid points). Can authors comment on this aspect?*

Thank you for the remark. The spatial resolution of the wind fields is 17 km. It has been corrected in the paper.

We agree that a spatial resolution of 600 m could not be enough for solving wave breaking and shallow water processes along the whole coastline. However, such a resolution can provide a good assessment on wave generation and propagation (please refer to the error metrics in Table 1 and Figure 5.

The use of unstructured meshes avoids nesting, that may be an important source of uncertainty. This work shows that the continental shelf (mainly, the inner shelf) joint with the wind fields patterns, are strongly correlated with the wave fields.

Additionally, we have previously dealt with this issue by carrying out an inverse distance weight type of interpolation in order to gain resolution, before computing the geo-statistical anisotropy of Vw and Hs.

*If you define the resolution in meters, I guess it is smaller near the coast, not higher.*
We have substituted "higher" by the term "denser", in order to make the text clearer to the reader.

*Which period is spanned by the analysis? (February 2017? Why not using a longer period, say 2016-2017?).*

The period ranges from November 2016 to March 2017. Such limitation comes from the availability (at the moment of writing) of the wind fields.

*As far as waves are concerned, using the proposed methodology, the distance of the border should change between August (instead of November, Fig. 9) and February. A season-based classification of the border would be a sound improvement of the paper.*

As mentioned above, unfortunately, the authors do not have data for August.

We agree that the anisotropy of the wind and wave fields may depend on seasonality. However, according to our data, the 90th percentile of the variance of the anisotropy does not depend on the seasonality. We would like to make an emphasis that it is the quantile, and not the absolute value, that stays the same throughout the year.

*About the radius and quantile threshold. Those two values seem to me quite arbitrary and not directly physically-based. Authors could do a sensitivity analysis to show how the results depend upon those values (the quintile, in particular). This is an important task in order to evaluate the stability of the proposed methodology (and then make it usable in other contexts) and provide the uncertainty of the location of the boundary. o It would be useful to plot the quantile values on the heatmaps.*

The 90th percentile in an environmental parameter (e.g. Hs) is a convention commonly followed in Literature (Eastoe 2013, Bernadara 2014). We have used this same idea, applied to the variance of the geo-statistical anisotropy of Vw and Hs. This concept is illustrated on Figs. 8 and 9.

*Please explain what is the count in the heatmap's caption?*
The comment "The counts are the number of elements within a hexagon.
" has been added to Figs. 6 and 7.

Additionally, the following text has been added (pp. 7, lines 7-11): Heatmaps are used to represent the spatial distribution of the geo-statistical anisotropy, showing how the density of R behaves as a function of distance to the coast and time (see Figs. 6 and 7). These maps are scatter plots that act as a 2D-histogram, in which two variables (in this case, R and distance to the coast) are grouped in pre-defined intervals. The elements selected to aggregate samples for the heatmap are hexagons with side 5km and a scale for anisotropy of 20 units for both R(Vw) and R(Hs) .

*I suggest putting the wave model assessment in a dedicated sub-section*

***separated from the results of the anisotropy analysis.***

The appearance of the validation graphs along with the text for the anisotropy analysis is because of the limitations of the LaTeX file. We are certain that the figures will appear along with the text, in the final version.

***Is there a reason why in the panels of Fig 8 and 9 the dashed green vertical lines are not aligned (horizontal axes seem consistent)?***

The figures are about 1 mm narrower above because of the number of digits in the y-axis. We believe that it should not significantly interfere in the interpretation of the graphs.

---

## Author Comment (AC2) · 9 Jan 2019

**Reviewer #2**

*Review: "The land-sea coastal border: A quantitative definition", by Sánchez-Arcilla et al.*

*Recommendation: major revision.*

*Summary: The authors attempt to provide a quantitative and generalizable definition of "land-sea" zone, i.e., cross-shore width of that particular marine area that is strongly affected by the presence of the continent. The methodology is based on the measure of anisotropy of specific, vectorial and/or scalar fields of environmental parameters. For this work, the authors use wind velocity and significant wave height from well-calibrated and validated numerical outputs.*
*---*

*General comments*

*The work the authors present is really intriguing and I particularly like the idea of defining a "coastal zone" by using environmental variables in a quantitative fashion. However, while the specific variables the authors consider in this application (i.e. wind velocity and significant wave height) are particularly suitable for the study area they might not work for a different environment, where, for instance, wind and wave patterns do not actually characterize the coastal zone. As stated by the authors, river plumes or, more in general, bio-geochemical processes may lead for a better definition of a "coastal area" and, as a consequence, the methodology here proposed might not be suitable. All this is at the base of my main criticism:*
*to state that such a methodology provides a "quantitative definition for the land-sea (coastal) transitional area" is too strong; although I like generalizations, I still believe that a "land-sea (coastal) transitional area" can be defined by starting from the specific physical, and/or biogeochemical, and/or geological, and/or ecological process we want to investigate.*

(pp. 5, lines 8-12) The following definition has been added to the text: "Although other definitions of the coastal boundary can be based on river plumes or bio-geochemical processes, it has been intended to focus on a more hydro-dynamical expression of such boundary for wave-driven coasts."
We have also specified through the text (Abstract, Introduction and pp.5) that we only focus on wave-driven environments.

*A second comment regards the poor connection between the pure mathematical/statistical part and the environmental application. I would have*

*appreciated a better explanation of the statistics by starting from the environmental data, also discussing physical meanings and assumptions. To present the theoretical background as it is leaves the reader with some doubts regarding the feasibility of the methodology.*

(pp.5, lines 5-8) It is intended to show that, as one approaches the coast, the wind and the wave fields should present a higher geometric anisotropy, that is, they should present predominant wind and wave directions. Furthermore, there should be a geo-statistical boundary to the value of this anisotropy that could help define a coastal boundary.

**Specific, minor comments**

**Abstract - replace "perpendicular" with "cross-shore" in line 2**
The suggested change has been carried out.

**Introduction - The are several definition of what a Land-sea border is (Shaw et al., 2008; Geleynse et al., 2012). I would avoid (at least, at the beginning) to frame the land-sea border within this specific definition. Instead, it would be better to state that over land-sea border areas occur specific met-ocean dynamics that actually characterize land-sea coastal border. The aim of this work is to quantitative define the extension of this area. (see general comment).**

We agree with this. Hence, the definition by Wright is obviated from the "Introduction".

Thank you very much for the references. The text has been revised as follows:

*"There is, thus, a need for a systematic and objective definition of the coastal fringe that considers underlying processes and that has general applicability allowing for the time/space dynamics of this fringe. This type of approach has been explored in the literature, where for instance Sánchez-Arcilla and Simpson (2002) reviewed a number of possibilities based on a dynamic balance of competing processes (i.e. drivers) such as inertial effects, geostrophic steering, sea bed friction or water column stratification. Another suitable option is to focus on the consequences of such processes, such as the nearshore morphodynamic features (Geleynse et al., 2012) (i.e. deltas, sand spits, overwash fans, beach berms). Both complementary classifications requires spatial data that needs to be updated accordingly within timescales that may range from years (i.e. long-term erosion due to sea level rise) to days (i.e. storm-scale)."*

**- "Sentinel data" (in line 3-pag 2) ; the general reader might not be familiar with**

*the sentinel missions and, therefore, might not understand that here authors are referring to satellite data. Please, introduce the Remote Sensing approach properly.*

This version does not put so much emphasis on the Sentinel satellites, but rather on wave altimeter data in general.  The following sentences replace the original lines 3 and 4: "The recent advent of high resolution and short revisit time provided by them offer an alternative source of information for such a coastal zone definition although with some limitations since the data may start degrading at a few kilometres (order 10km) offshore from the coast (Cavaleri and Sclavo, 2006; Wiese et al., 2018; Cavaleri et al., 2018)."

*- "Because of that" (in line 5-pag 2); Please, be more specific. It's not clear the use of Sentinel data in defining land-sea limits and what the authors mean with degradation of data. "necessary"; too strong, I would write "useful" rather than necessary.*

The following sentences replace the original lines 5 to 8: " The land boundaries induce error in the satellite observations. Hence, it is useful to use high resolution numerical simulations supported by in-situ data so that land-sea boundary effects are properly captured for the subsequent coastal definition that will be based on the inhomogeneity introduced by the presence of the land boundary."

*- "coastal anisotropy" (in line 13-pag 13); I would write "anisotropy of environmental parameters" rather than coastal anisotropy*

The suggested change has been carried out. The following sentences replace the original lines : "The aim of this paper is to analyse the geo-statistical anisotropy of nearshore wind and waves, in wave-driven coasts. From that, what follows is to propose a new quantitative and objective definition for the land-sea border that benefits from these high-resolution (spatial and temporal) fields and from the underlying process-based knowledge. This definition can be useful to determine a set of criteria for numerical purposes (e.g. nesting coastal domains) but also for more practically oriented applications (e.g. offshore limit for outfall dispersion)."

*Theoretical background - G(x) in line 10 should be G(y), as far as I am missing something;*

The recommended correction has been carried out.

*As I suggest in the General Comments, this section would be much clearer (and the ms much stronger) if the theoretical background is explained by starting from environmental variables. As it is, the reader might get confused.*

Thank you so much for the recommendation. We would like to leave the explanation with environmental variables to the Methodology. The Theoretical background is intended to be an introduction of the mathematical tools used.

***Study area - By reading the section it comes natural to think that the analysis is particularly suitable for this study area, thus difficult to generalize***

It is intended to propose this methodology. The proposed limit to the coastal fringe is not to be generalized, but the methodology can help find the indicated one for each coast.

***References***

***Shaw, J. B., Wolinsky, M. A., Paola, C., & Voller, V. R. (2008). An image based method for shoreline mapping on complex coasts. Geophysical Research Letters, 35(12).***

***Geleynse, N., Voller, V. R., Paola, C., & Ganti, V. (2012). Characterization of river delta shorelines. Geophysical Research Letters, 39(17).***

---

## Author Comment (AC3) · 9 Jan 2019

**Comment**

*The authors present a methodology for determining the land-sea transitional area based on the empirical distribution of anisotropy in meteorological and ocean processes. This is an interesting article, however it will be beneficial for the audience if the authors could provide some feedback on the following matters:*

*1 Definition of anisotropy*
*In [Chorti et.al., 2008] a non-parametric estimator of statistical anisotropy was proposed, for which an approximate estimate of the anisotropy statistics distribution was provided in [Petrakis et.al., 2017]. While the authors cite [Chorti et.al., 2008], from the rest of the references it is not clear if anisotropy is defined as in geostatistics (statistical anisotropy: directional dependence of correlation functions) or as in (geo)physics (directional variation of a physical property, e.g., elasticity, permittivity). Also it is not clear how anisotropy is estimated. The authors should clarify, by providing the definition of anisotropy and the estimator they use.*

It is very true that we should specify that it is a geo-statistic anisotropy (geometric). Therefore, we have specified at the aim of the paper: "The aim of this paper is to analyse the geo-statistical anisotropy of nearshore wind and waves, in wave-driven coasts. From that, what follows is to propose a new quantitative and objective definition for the land-sea border that benefits from these high-resolution (spatial and temporal) fields and from the underlying process-based knowledge."

*2. Spatial resolution of wind and wave fields*

*For both fields there are sub-domains with anisotropy ratio estimates of*
*R≈100 or more. Therefore, the largest correlation length within such sub-domains is larger by two orders of magnitude compared to the smallest correlation length over the perpendicular principal axis. Assuming stationarity, for an accurate estimation of anisotropy a field should be sampled at a sufficiently large domain, to satisfy ergodicity, and at a high resolution, in order to capture the spatial variability at length scales below the smallest correlation length. The authors estimate anisotropy over circular sub-domains of 5km*
*radius. Some representative field maps would be useful to justify that the sub-domains are sufficiently large and contain an adequate number of measurement samples for the fulfillment of the aforementioned requirements.*

We have modified the flow-chart to clarify that we have interpolated the wind and the wave fields in order to have enough resolution to obtain the anisotropy.

(pp. 7, lines 1-2) Also, we have reorganized the text so these lines say: "The geo-statistical Anisotropy needs to be computed on a regular grid and therefore, both wind velocity ($V_w$) and significant wave height ($H_s$) have been interpolated on a rectangular mesh, first on a grid of 1 km then to a finer mesh of 10m."

**References**

Chorti, A., Hristopulos, D. T. (2008). Nonparametric identification of anisotropic (elliptic) correlations in spatially distributed data sets. IEEE Transactions on Signal Processing, 56(10 I), 4738-4751. doi:10.1109/TSP.2008.924144

Petrakis, M.P., Hristopulos, D. T. (2017). Non-parametric approximations for anisotropy estimation in two-dimensional differentiable gaussian random fields. Stochastic Environmental Research and Risk Assessment, 31(7), 1853-1870. doi:10.1007/s00477-016-1361-0

---

## Author Comment (AC4) · 9 Jan 2019

*Reviewer #3*

*GENERAL COMMENTS:*
*The paper addresses a relevant scientific question which is well within the scope of Ocean Science. The authors present a methodology with the aim to quantitatively define the land-sea boundary in wave-dominated and micro-tidal environments. The presented methodology builds on met-ocean datasets which are well-known and frequently used in the field. The authors conclude that the proposed land-sea boundary (coastal fringe) definition is a generic method. However, as also stated by the author, the correct choice of met-ocean or biogeochemical variables might be case dependent and the presented application is tailored to the case specific conditions at the Catalan coast.*
*---*

*It would significantly improve the general applicability of the method if the authors could briefly describe how one should choose the variables that reflect the influence of the land border in a specific application.*
(pp. 5, lines 5-8) The following definition has been added to the text: "Although other definitions of the coastal boundary can be based on river plumes or bio-geochemical processes, it has been intended to focus on a more hydro-dynamical expression of such boundary for wave-driven coasts."

*Moreover, by comparing the results to other land-sea border definitions (validation) and by providing uncertainty estimate of the computed coastal zone limit the authors would make the methodology stronger.*

Thank you very much for the remark. We have added several comments on this issue throughout the paper. Please find two examples below:

(pp. 2, lines 1-8) *"There is, thus, a need for a systematic and objective definition of the coastal fringe that considers underlying processes and that has general applicability allowing for the time/space dynamics of this fringe. This type of approach has been explored in the literature, where for instance Sánchez-Arcilla and Simpson (2002) reviewed a number of possibilities based on a dynamic balance of competing processes (i.e. drivers) such as inertial effects, geostrophic steering, sea bed friction or water column stratification. Another suitable option is to focus on the consequences of such processes, such as the nearshore morphodynamic features (Geleynse et al., 2012) (i.e. deltas, sand spits, overwash fans, beach berms). Both complementary classifications requires spatial data that needs to be updated accordingly within timescales that may range from years (i.e. long-term erosion due to sea level rise) to days (i.e. storm-scale)."*

(pp. 16, lines 13-17) We have added the comment: *"The coastal boundaries suggested by Sánchez-Arcilla and Simpson (2002) for the Catalan Coast can be 0.1-0.6km (frictional coupling of fluids between shelf and nearshore), 10km (non-linear coupling between shelf and slope), 1km (non-linear coupling between shelf and nearshore), among other suggested values of the same order of magnitude. The "l" provided in this analysis is slightly larger than the value given for the frictional coupling of fluids between shelf and nearshore, whereas it is similar or smaller than in the non-linear couplings. Nevertheless, the orders of magnitude are similar."*

**The scientific methods and assumptions are in general valid and clearly outlined, even though further clarifications are required at certain sections (see specific comments). The paper is well structured in general; however, certain elements should be better explained (see specific comments).**

**SPECIFIC COMMENTS:**
**Title:**
**- The title should indicate that the methodology to quantitatively define the land-sea coastal border was only tested for a case study in a wave-dominated and micro-tidal environment.**

The new title is adapted to this idea: "The land-sea coastal border: A quantitative definition by considering the wind and wave conditions in a wave-dominated, micro-tidal environment".

**Abstract:**
**- I propose to change the term "90th quantile" to "90th percentile" throughout the paper. The authors refer to the 90th 100-quantile which is called percentile.**

Thank you very much for this remark. Therefore, we have substituted "quantile" by "percentile".

**Study area:**
**- The authors state (page 4, line 5) that the focus area is the Spanish north-eastern Mediterranean coast due to the availability of in-situ and Sentinel images for support. It is not clear how the Sentinel images were utilised in the methodology as a support (unless they were used for the SWAN model validation, which is not stated in the paper).**

We agree with this point. Please refer to the answer below, to the same referee, referring to Fig. 5.

**Methods:**
**- Further background information on the Unified Model and/or the wind field data should be given. The wave data is explained in much more detail**

***compared to the wind data.***

(pp. 6 lines 4-10) The following explanation has been added to the text: "There are two atmospheric prognostics: the dry one (three-dimensional wind components, potential temperature, Exner pressure and density) and the moist one (specific humidity and prognostic cloud fields (Walters et al. (2011)). Both long and short radiations (from the sun and the Earth itself) are included, whereas the effect of aerosols reflecting them is taken into consideration."

*- According to Cullen (1993) the operational forecast grid for the Unified Model is 0,833 degree (latitude) and 1,25 degree (longitude), whereas the standard climate and upper atmosphere configuration uses 2,5 degree (latitude) and 3,75 degree (longitude). Is it really the second configuration which is used in this paper? This resolution would mean approximately 250km (latitude) and 310 km (longitude). That is a very coarse resolution for this purpose.*

Thank you very much for this remark. The horizontal resolution of the atmospheric model was a gridsize of 17 km, the same that the UK Met Office global deterministic forecast model. (pp. 6, 12-13) "The computational domain of the wind field spans the whole Mediterranean Sea using a regular grid with spacing of 17km and a time step of 1h."

Also, we have improved the flow chart to clarify, along with the existing definition of the methodology, that we interpolate the wind/wave data field in order to obtain a finer grid from which to compute the geo-statistical anisotropy along the transects.

*- I suggest adding steps to the methodology figure (Figure 3) for the wave and wind model validation, interpolation, as well as for the distribution fitting (Gaussian copula model).*

We have proceeded as indicated. The caption of the figure is also modified: "Flow-chart summarizing the methodology used in this paper. The dashed blue rectangle represents the input data, the red dashed rectangle indicates the output data. Only the wind velocity is obtained from an external source, the rest of the steps have been carried out for this analysis. Rectangles indicate data generation (input/output) and rhombuses the subsequent analyses of the proposed methodology."

*- It is mentioned (page 6, line 9) that wave fields have been validated. Validation results should also be included for the wind field data. Reference to the wind field validation is only given in the discussion section (page 15, line 3-4). I suggest moving this sentence to the Methods section where the United Model is described.*

Validation of the wind fields is not included in this paper. Then, we have added the following: (pp. 5 line 26) "These wind data are validated in (Martin et al. ((2006))."

*- Is there any reason why the 90th percentile is used in equation 6? Is it based on expert knowledge or literature?*

Indeed, the selection of the 90th quantile is a convention commonly followed in Literature (Eastoe 2013, Bernadara 2014).

*Results:*
*- The figures 6,7,8,9 are presented on pages 11-to 14 while described on page 7. This makes it hard for the reader to follow the paper. Consider to move them closer to the place where they are described.*

We agree with this suggestion. The figures appear after the text, in the source file. This problem happens because the graphs are large and self locate in these pages. We believe that this phenomenon would only occur in this pdf format, but it should be automatically solved in an online edition.

*- In Figure 5 red dots are labelled as Altimeter data. Is this data coming from Sentinel images? If yes, please explain both in the legend and in the text, and also add which mission it is (e.g. 3A).*

The altimeter data comes from Jason-2, Jason-3 and Cryosat. Sentinel 3A data has not been used in this contribution.

(Fig. 5) We have added the clarification:  "The red dots are altimeter data from altimeter data (Jason 2, Jason 3 and Cryosat)"
(pp. 7 line 18) We have added: "The SWAN model simulations have been validated with significant wave height ($H\_s$), registered with buoys and altimeter data, at the southern (Tarragona location) and northern (Begur location) coastal sectors (Figs. 4 and 5)."

*- The calculated coastal zone limit values are not mentioned explicitly in the results section, even though they are depicted in Figure 8-9 and mentioned in the abstract. I suggest mentioning them in the text as this is the main objective of the methodology.*

We have modified the text so now it reads: "The coastal zone limit "l", corresponding to the 90th percentile of the total variance (fringe between 0
and 100km), is calculated from equation 6 (Figs. 8 and 9) and is 3km. It is consistent with time interval (month of study) and location (sector)"

*- Please use the word "coastal zone limit" consistently. Sometimes it is only called "limit".*

The suggested action has been carried out throughout the text ( pp.10 line 1, pp. 17, line 5).

***Discussion:***
***- The authors write that "The calculated anisotropies should be as robust as the starting wave or wind fields that are employed in the analysis" – that is why the robustness of the wind field should be better defined in the Methods section.***

We agree with the reviewer. The UK Met Office wind fields has shown systematically good accuracy (see Martin et al. 2006, Brown et al. 2012, Walters et al. 2011). We have assumed that the wind fields have state-in-the-art accuracy and we have focused on validating the wave fields in the Results and we hope that this could be a valid procedure.

***- Figure 6-7: the description of the hexagons in the heatmap should be added to the figure as they are only described in the text. Also, the description of the blue dashed line at 20 km should be added as in Figure 8 and 9.***

The suggested changes have been performed.

***- Figure 10: The x axis represents the months within a year cycle. Which year is it? And why only months 1, 2, 3, 11,12 were selected?***

These months (year 2016; 11, 12 and year 2017; 1,2,3) span the available data. As mentioned above, the available wind fields ranged this timeline (at the moment of writing this paper). We accept this shortcoming as a limitation of our contribution.

The following text has been added to the caption of the figure: "The plot shows the variation with time (horizontal axis) between November of 2016 and March of 2017. The parameters are placed in a manner that they start from January."

***- The authors write (page 16, line 5) that the correlation between R_Vw and R_Hs is the strongest for the Begur transect. On the other hand, in Figure 10 the Begur transect (orange dots) has a correlation parameter around 0 (max ~ 0.026). This figure indicates that the Mataro transect has the strongest correlation parameter, not Begur.***
***I suggest clarifying this.***

Thank you for pointing this out. We have clarified in the text that it is the "overall" dependence that is stronger in Begur: "The overall mutual dependence of R_vw and R_Hs is strongest for the northern-most transect (Begur), where the topo-bathymetric control of the Pyrenees and their submerged signature becomes better defined."

***References:***
***- The number of references is rather high (57). Moreover the share of references originating from the same authors is also high.***

Although all references are of strong interest, we have followed the suggestion of the referee to reduce the number of references. Here is a list:

-Bolaños and Sánchez-Arcilla (2006), as it can be represented by Bolaños et al. (2009).

-Bolaños et al. (2007), as it only appears once in the text and along other references.
-Pallarés et al. (2013), for the same reason.
-The thesis of E. Pallarés can be represented by Pallarés et al. (2014).
-Sánchez-Arcilla et al. (2008), as it is similar to Bolaños et al. (2009).
-Sánchez-Arcilla et al. (2016), as it only appears once, and along with another reference. Also, it is more about ports.
-Sierra et al. (2017) has been obviated, as it is well represented by the other bibliography that accompany it in the "Introduction".

Additionally, it has been added new references, that the authors consider that suit better the general messages of this contribution. Note that some of these references come from the other reviewers' suggestions.

**REFERENCES**

Bernadara, P., Mazas, F., Kergadallan, X. and Hamm, L. (2014). A two-step framework for over-threshold modelling of environmental extremes. Natural Hazards and Earth System Sciences, 635--647.

Brown, A., Milton, S., Cullen, M., Golding, B., Mitchell, J., and Shelly, A.: Unified modeling and prediction of weather and climate: A
25-year journey, Bulletin of the American Meteorological Society, 93, 1865–1877, 2012.

Eastoe, E. , Kouloulas, S. and Jonathan, P. (2013). Statistical measures of extremal dependence illustrated using measured sea surface elevations from a neighbourhood of coastal locations. Ocean Engineering, 68--77.

Martin, G. M., Ringer, M. A., Pope, V. D., Jones, A., Dearden, C., and Hinton, T. J.: The physical properties of the atmosphere in the new
Hadley Centre Global Environmental Model (HadGEM1). Part I: Model description and global climatology, Journal of Climate, 19,
1274–1301, 2006.

Walters, D. N., Best, M. J., Bushell, A. C., Copsey, D., Edwards, J. M., Falloon, P. D., Harris, C. M., Lock, A. P., Manners, J. C., Morcrette,
C. J., et al.: The Met Office Unified Model global atmosphere 3.0/3.1 and JULES global land 3.0/3.1 configurations, Geoscientific Model
Development, 4, 919–941, 2011.